

# Temporal changes in snow albedo, including the possible effects of red algal growth, in northwest Greenland, simulated with a physically based snow albedo model

Yukihiko Onuma[1], Nozomu Takeuchi[2], Sota Tanaka[2], Naoko Nagatsuka[3], Masashi Niwano[4], and Teruo Aoki[3,4]

[1]Institute of Industrial Science, University of Tokyo, Chiba, 277-8574, Japan
[2]Graduate School of Science, Chiba University, Chiba, 263-8522, Japan
[3]National Institute of Polar Research, Tokyo, 190-8518, Japan
[4]Meteorological Research Institute, Japan Meteorological Agency, Tsukuba, 305-0052, Japan

*Correspondence to*: Yukihiko Onuma (onuma@iis.u-tokyo.ac.jp)

**Abstract.** Surface albedo of snow and ice is substantially reduced by inorganic impurities, such as aeolian mineral dust (MD) and black carbon (BC), and also by organic impurities, such as microbes that live in the snow. In this paper, we present the temporal changes of surface albedo, snow grain size, MD, BC, and snow algal cell concentration observed on a snowpack in northwest Greenland during the ablation season of 2014 and our attempt to reproduce the changes in albedo with a physically based snow albedo model. We also attempt to reproduce the effects of inorganic impurities and the red snow algae (*Chlamydomonas nivalis*) on albedo. Concentrations of MD and red snow algae in the surface snow were found to increase in early August, while snow grain size and BC were found to not significantly change throughout the ablation season. Surface albedo was found to have decreased by 0.08 from late July to early August. The albedo simulated by the model agreed with the albedo observed during the study period. However, red snow algae exerted little effect on surface albedo in early August. This is probably owing to the abundance of smaller cells ($4.9 \times 10^4$ cells L$^{-1}$) when compared with the cell abundance of typical red algal snow ($\sim 10^8$ cells L$^{-1}$). The simulation of snow albedo until the end of the melting season, with an algal growth model, revealed that the reduction in albedo attribute to red algae could equal 0.004, out of a total reduction of 0.102 arising from the three impurities on a snowpack in northwest Greenland. Finally, we conducted scenario simulations using the snow albedo model, coupled with the algal growth model, in order to simulate the possible effects of red algal blooming on snow albedo under warm conditions in northwest Greenland. The result suggests that albedo reduction by red snow algal growth under warm conditions (surface snow temperature of +1.5°C) reached 0.04, equivalent to a radiative forcing of 7.5 W m$^{-2}$ during the ablation season of 2014. This coupled albedo model has the potential to dynamically simulate snow albedo, including the effect of organic and inorganic impurities, leading to proper estimates of the surface albedo of snow cover in Greenland.



## 1 Introduction

30        The Greenland Ice Sheet, which is the largest continuous body of ice in the Northern Hemisphere, has been losing mass rapidly since the 2000s (Rignot et al., 2008). The increase of in the melting of snow and ice is likely to be caused by reduction of surface albedo as well as temperature rise (Tedesco et al., 2011; Box et al., 2012). Therefore, it is important to understand the physical processes causing the reduction in albedo, and to estimate current and future snow and ice albedo accurately on 35   the Greenland Ice Sheet.

        Surface albedo plays an important role in the balance of energy over the snow surface. Snow albedo is approximately 0.9 for fresh snow and gradually decreases to approximately 0.5 for granular snow during the melting season (Wiscombe and Warren, 1980). Because a reduction of snow albedo increases the absorption of solar radiation by a snowpack, the reduction in albedo accelerates the melting of snow. The major factors affecting surface albedo are snow grain size and abundance of 40   light-absorbing impurities in the snow (Warren and Wiscombe, 1980; Aoki et al., 2011). An increase in the amount of snow impurities and in the snow grain size cause more absorption of solar radiation in the visible (300–700 nm) and near-infrared (700–3000 nm) regions (Warren and Wiscombe, 1980; Aoki et al., 2000). Major light absorbing impurities in a snowpack are black carbon (BC), which is derived from the combustion of fossil and solid fuels and from biomass burning (Bond et al., 2013), and mineral dust (MD), which is transported by wind from local or distant arid terrestrial surfaces (Bøggild et al., 2010). 45   For example, a mass concentration of 10 ppbw of BC in wet snow can reduce the albedo by 0.01 (Warren and Wiscombe, 1985). Although light-absorption by MD in the visible region was reported to be lower by 0.7% of that by BC (Aoki et al., 2011), the mass concentration of MD was ten times greater or more than that of BC in the snowpack in the Greenland Ice Sheet. Thus, the impact of MD on snow albedo cannot be ignored (Steffensen, 1997). In addition, organic carbon (OC) is an impurity that absorb light in the visible spectrum (Kirchstetter et al., 2004; Andreae and Gelencsér, 2006). OCs in atmospheric 50   aerosols, consisting of burned fossil fuel, plant materials, viable microbes (bacteria, viruses, and fungal spores), soil organic matter, and marine aerosol (Jacobson et al., 2000; Cerqueira et al., 2010) might be present surface snow and reduce its albedo.

        Physical models of snow surface albedo have been developed to calculate the surface albedo of snow containing various impurities. Physical models can reproduce snow albedo as a function of snow grain size, impurities (BC and MD), and direct and diffuse solar radiation (Wiscombe and Warren, 1980). Such models have been developed in recent years. For example, 55   Flanner and Zender (2005; 2006) proposed a multilayer snow albedo model, which is incorporated into a land surface model in a general circulation model, in order to simulate the microphysics and radiative properties of snow at a global scale. Physically based snow albedo model (PBSAM) developed by Aoki et al. (2011) separately calculates broadband albedos for the ultraviolet/visible (200–700 nm) and near infrared (700–3000 nm) wavebands, at the snow surface, taking into account the spectral radiation properties of impurities (BC and MD) in those spectra. This model can efficiently simulate snow albedo 60   using a look-up table, which consists of an albedo dataset calculated with a radiative transfer model for various environmental variables (snow grain size, snow water equivalent, snow impurity concentration, and solar zenith angle). Global climate simulation (e.g. of surface net shortwave flux) using an Earth system model with a snow albedo module, including light-



absorbing impurities (BC, MD and OC), suggests that the contribution of aerosol OC to total visible absorption in the snow surface was smaller than that of BC and MD (Yasunari et al., 2015).

Although many albedo models have been developed to include inorganic impurities (BC and MD) and aerosol organic impurity, recent studies have suggested that microbes growing in snow, such as snow algae, also affect snow albedo (Yallop et al., 2012; Takeuchi, 2013; Aoki et al., 2013; Lutz et al., 2016; Cook et al., 2017a; Cook et al., 2017b). Snow algae are cold-tolerant photosynthetic microbes growing on snow and ice and are commonly found globally on glaciers and snowfields. Blooms of snow algae occur on thawing snow surfaces and change the color of snow to red or green (Thomas and Duval,

1995; Hoham and Duval, 2001; Takeuchi et al., 2006). Red colored snow results from a bloom of snow algae, which are typically *Chlamydomonas (Cd.) nivalis*, and can be observed widely in polar and alpine snow fields (Hoham and Duval, 2001; Segawa et al., 2005; Takeuchi, 2013; Hisakawa et al., 2015; Lutz et al., 2016; Tanaka et al., 2016; Ganey et al., 2017; Segawa et al., 2018). Many observational studies have reported the quantitative effect of the algal blooming on snow albedo. For example, algal blooming has a potential to reduce snow albedo by 0.13 on Arctic glaciers (Lutz et al., 2016). Because the

impact of snow algal blooming on albedo is comparable to those of BC and MD, albedo models including the effect of snow algae have been established recently. For example, a bio albedo model proposed by Cook et al. (2017a; 2017b) can simulate spectral albedo using calculations of radiative transfer that incorporate biological variables (cell concentration, cell size, and pigment composition in cell), snow physical properties (specific surface area, density, and layer thickness), and irradiance. This model satisfactorily reproduced the spectral albedo of algal blooming snow and temporal change of surface albedo on a

glacier in Greenland. However, the physical properties and inorganic impurities (BC and MD) in the snow used in the simulation were not representative field values, but assumed constant values based on previous studies at other sites of Greenland. Thus, the effect of snow algae on intact surface albedo has yet to be quantitatively assessed. In addition, temporal changes in algal abundance were not used at the model calculation. As snow algal abundance can change significantly because of their growth over time, snow albedo simulations should incorporate a growth model of snow algae.

In this study, we aimed to reproduce the temporal changes of snow albedo observed on a snowpack during the melt season of Qaanaaq Glacier in northwest Greenland. We used the PBSAM and included the effects of snow algae, as well as inorganic impurities (BC and MD). Snow physical properties (snow grain size, temperature, density, and spectral reflectance in the visible band) and the abundances of the three impurities (BC, MD, and snow algae) in the surface and subsurface snow were periodically quantified in the snowpack from June to August in 2014. The PBSAM was updated to incorporate the effect of

snow algae, and temporal changes of the surface albedo were calculated by using the model and incorporating the observed meteorological conditions, snow physical properties, and the three impurities. The impacts of the observed and simulated snow algal blooming on snow albedo were quantified using the updated PBSAM. In addition, we simulated temporal change in snow algal abundance by using a snow algal growth model to reproduce the change in albedo throughout the entire melting season.



## 2 Method

### 2.1 Study sites and observation methods

Filed investigations were conducted at the Qaanaaq Ice Cap in northwest Greenland (Fig. 1) from June to August in 2014. The Qaanaaq Ice Cap, which lies on a small peninsula in northwest Greenland, covers an area of 286 $km^2$ and has an elevation of approximately 1110 m a.s.l. (Takeuchi et al., 2014; Sugiyama et al., 2014). We selected a study site close to the SIGMA-B automatic weather station (AWS) (77°32' N, 69°04' W, 944 m a.s.l., Aoki et al., 2014a) on the ice cap, which is easily

accessible on foot from the Qaanaaq village. The site is located near the equilibrium line of the glacier, which was 1001 m a.s.l. in 2014, as determined by a mass balance study (Tsutaki et al., 2017). As reported previously, snow algae visibly bloomed on the snowpack at the study site from late July to August in 2014. They consisted mostly of the spherical red cells of *Cd. nivalis*, and their mean diameter was $21.3 \pm 2.3$ µm (Onuma et al., 2018).

The meteorological conditions considered in this study were measured at the SIGMA-B AWS, which was installed in 2012

as part of the project Snow Impurity and Glacier Microbe effects on abrupt warming in the Arctic (SIGMA, Aoki et al., 2014a). Air temperature and radiation fluxes of the upward and downward shortwave, and upward and downward longwave were collected hourly from June to August 2014 using the AWS. The temperature sensor and pyranometers of the AWS were placed at heights of 3.0 and 2.5 m above the snow surface, respectively. Surface albedo was calculated from the ratio of upward to downward shortwave radiation. The surface albedo was corrected to include the effect of local slope (four degrees) on snow

albedo, according to Jonsell et al. (2003). The time used in this study was Greenland local time (LT), which is 2 hours later than Greenwich Mean Time. Detailed settings of the other meteorological sensors have been described by Aoki et al. (2014a).

Collection of snow samples and measurement of snow properties were periodically conducted at the site from day 168 (June 17, 2014) to 215 (August 3, 2014) to obtain the level of abundance of the impurities and the physical properties of the surface snow. The optically equivalent snow grain size was measured with a handheld lens according to Aoki et al. (2007). Snow

temperature was measured with a thermistor sensor (CT-430WP, Custom Ltd, Tokyo, Japan). The snow density was measured using a density sampler. Variation of snow layer thickness at each observational date relative to that on day 168 was defined as the relative snow surface level for estimating snow melting. Spectral reflectance of the snow surface in the visible wavelength range (350–700 nm) was measured with a portable spectroradiometer (MS-720, Eiko Seiki, Japan). Surface snow was collected after the measurement to quantify the impurity content.

Snow samples were collected from one to three surfaces selected randomly at the study site. They were collected from two snow layers, at depths of 0 to 2 cm (surface) and 2 to 10 cm (subsurface), using a stainless-steel spatula. At each layer, snow samples were collected separately to quantify the algal cell, MD particles, and OC and BC contents. The amounts of snow sample used for algal cell analysis ranged from 22 to 36 g for the surface and 21 to 33 g for the subsurface. Samples for analysis of MD particle and BC and OC concentrations ranged from 500 to 2200 g for the surface and 500 to 2100 g for the subsurface.



## 2.2 Quantification of snow impurities

Mass concentrations of MD in the snow were quantified by the combustion method (Takeuchi and Li, 2008; Onuma et al., 2018). Snow samples were collected at the site in dust-free plastic bags from the surface and subsurface, as described in the previous section. These samples were melted at room temperature in Qaanaaq village and their mass was measured with a weight scale. The dust precipitated in the bag was preserved in 30 mL clean polyethylene bottles, which were then transported to Chiba University, Japan, for analysis. The samples were dried (60°C, 24 h) in pre-weighed crucibles and then combusted (500°C, 3 h) in an electric furnace to remove organic matter. The mass of mineral particles per melt water volume (mg L$^{-1}$) was obtained from the combusted sample weight and sample volume, since only mineral particles remained after combustion. Mass concentrations of MD in surface snow at the study site during the observational period have been observed in previous studies (Onuma et al., 2018).

Mass concentrations of OC and elemental carbon (EC) in snow were quantified from filtered snow samples by the thermal optical reflectance method (Chow et al., 1993; Kuchiki et al., 2015). Snow samples were collected in dust-free plastic bag from the snow surface and subsurface, as described in the previous section, at the site. The samples were melted at room temperature and pre-filtered through a 150 μm mesh filter to remove large particles, including leaves, insects, and clothing fibers. The melted samples were combined with ammonium dihydrogen phosphate (NH$_4$H$_2$PO$_4$) of 1.5 g per 100 mL$^{-1}$ as a coagulant and were then magnetically stirred and sonicated for 10 min. Previous studies have reported that the mean collection efficiency of a quartz fiber filter for BC particles increased to 95% when NH$_4$H$_2$PO$_4$ coagulant was added to the sample solution, compared with 5% efficiency without the addition of the coagulant (Torres et al., 2014). In addition, adding the NH$_4$H$_2$PO$_4$ coagulant to melted snow samples had no apparent effect on the measurement of OC by the thermal reflectance optical method (Kuchiki et al., 2015). The melted samples were filtered through a quartz fiber filter (pore size: 0.45 μm, 2500QAT-UP; Pall Corp., MI, USA) in a clean bench by atmospheric pressure. The filters were preserved in plastic cases before being transported to the Meteorological Research Institute in Tsukuba, Japan, for analysis. Mass concentrations of OC and EC were measured with an OC-EC Aerosol Analyzer (Sunset Laboratory Inc., OR, USA) using the thermal optical reflectance method. Samples were volatilized at 120, 250, 450, and 550°C in a pure helium atmosphere and then combusted at 550, 700, and 800°C in a 10% oxygen/90% helium atmosphere, in accordance with the Interagency Monitoring of Protected Visual Environments (IMPROVE) thermal evolution protocol (Chow et al., 2001). Throughout the analysis, laser reflectance from the filter deposit was continuously monitored to correct the OC pyrolysis. The filter reflectance usually decreased with increasing temperature in the helium atmosphere due to pyrolysis of organic material. When oxygen was added, the remaining light-absorbing carbon combusted, and the reflectance increased. The split point between OC and EC was identified by the timing of drastic change in reflectance. EC concentrations measured with the thermal optical method agree with the BC concentrations measured with an optical method using a particle soot absorption photometer to within 2% (Miyazaki et al., 2007). In the present study, we assumed that the component of EC was equal to BC in order to estimate the mass concentration of BC in snow. The mass



concentrations of OC and BC per melt water volume (mg L$^{-1}$) were obtained from the volatilization volume of carbon before and after the OC/EC split point, respectively. Kuchiki et al. (2015) provide a more detailed description of the method.

Abundance of snow algae was quantified by the direct cell count method (Takeuchi, 2013; Tanaka et al., 2016; Onuma et
al., 2018). Snow samples were preserved in Whirl-Pak® bags (Nasco, Fort Atkinson, Wisconsin, USA) and then melted in Qaanaaq village. The melted samples were preserved in 3% formalin in 30 mL clean polyethylene bottles before being transported to Chiba University, Japan, for analysis. Algal abundance was represented as algal cell concentration per unit melt water volume of the snowpack. Water samples of 50–1000 μL were filtered through a hydrophilized PTFE membrane filter (pore size 0.45 μm; Omnipore JHWP, Millipore, Japan), and the number of algal cells on the filter was counted two to five
times for each sample using an optical microscope (BX51, OLYMPUS, Japan), and cell concentrations (cells L$^{-1}$) were obtained from mean cell counts and filtered sample volumes. Cell concentrations for *Cd. nivalis* in surface snow at the study site have been published previously (Onuma et al., 2018).

## 2.3 Physically based snow albedo model (PBSAM)

A PBSAM was used to simulate snow albedo, including the effect of inorganic and organic impurities, in this study.
Broadband albedos in the snowpack were calculated by the PBSAM as functions of snow grain size and concentrations of impurities in a maximum of five layers of snow under solar illumination conditions (Aoki et al., 2011). In addition, PBSAM can calculate visible and near-infrared albedos using downward solar radiation in the visible and near-infrared regions, respectively. In order to include impurities of different optical properties (BC and MD in the case of Aoki et al., 2011), a snow impurity factor (SIF) was defined in the model. $SIF^i$ for sub-band of wavelength $i$ was calculated as follows:

$$SIF^i = k^{i,BC} C_{BC} + k^{i,MD} C_{MD} \tag{1}$$

where, $k^{i,BC}$ and $k^{i,MD}$ are the mass absorption cross sections (MACs) of BC and MD for sub-band $i$, respectively. MACs represent the absorption characteristic (absorption coefficient) for sub-band of different wavelengths (Table 1). $C_{BC}$ and $C_{MD}$ are the mass concentrations of BC and MD in snow, respectively. MACs and the mass concentrations of each impurity were used as the parameters and variables for simulation by PBSAM. Aoki et al. (2011) has further added the OC to equation (1) as
follows:

$$SIF^i = k^{i,BC} C_{BC} + k^{i,MD} C_{MD} + k^{i,OC} C_{OC} \tag{2}$$

where, $k^{i,OC}$ and $C_{OC}$ are the MAC and the mass concentration of OC for sub-band $i$, respectively. In their study, OC was assumed to be aerosol OC derived from the atmosphere. The MAC of the OC, as well as those of MD and BC, were assumed using an aerosol model (Hess et al., 1998), as reported by Aoki et al. (2011).
In the present study, red snow algae were included in the model as part of the OC. In order to convert the algal cell concentration (cells L$^{-1}$) into $C_{OC}$ (mg L$^{-1}$), a regression was applied to a scatter diagram of observed algal cell versus OC concentration of snow samples in this study. Because the MAC of snow algae is unlikely to be equal to that of aerosol OC, as has previously been used in PBSAM, we assumed the MAC for snow algae based on cell size and pigment composition (Table1). To calculate the MAC for snow algae, a log-normal size distribution (Hess et al., 1998, equation 3d) was assumed,





based on the measurement of algal cell size. The sizes of 100 *Cd. nivalis* cells were measured directly using Image-J for estimation of the normal distribution curve (mode radius = 11.4 µm, standard deviation = 1.18 µm). Because the effect of light absorption of snow algae on snow albedo should be calculated quantitatively in an albedo model, we calculated the imaginary part of refractive indices for *Cd. nivalis* according to Cook et al. (2017a, equation (2) and (3)). The imaginary part of refractive indices for the spectral region from 400 to 750 nm was calculated based on the pigment compositions (chlorophyll-a,

chlorophyll-b, primary carotenoids, and secondary carotenoids) that were assumed as the compositions of *Cd. nivalis* by them. The imaginary part of refractive index for non-absorption spectral regions by red snow algae, which are 200-400 nm and 750-3000 nm, was assumed to be that for pure water. The real part of the refractive index for the entire spectra from 200 to 3000 nm was assumed to be the same as pure water. The MAC for snow algae was calculated from Mie theory by assuming the spherical particles, using the log-normal size distribution previously calculated and the spectral refractive indices following

the protocol of Aoki et al. (2011).

Snow albedo was calculated with PBSAM, using $C_{BC}$, $C_{MD}$, $C_{OC}$, physical properties in surface (0–2 cm) and subsurface (2–10cm) snow and meteorological conditions recorded in this study. The observed thickness of the snow layer, snow density, and temperature and grain size were used as input data in PBSAM. Downward shortwave radiation measured at the AWS, the direct to diffuse insolation ratio, and the visible to near-infrared insolation ratio were also used as input variables. These two

ratios were calculated from observed downward shortwave radiation, upward and downward longwave radiation and air temperature at the AWS following the protocol of Niwano et al. (2012). Meteorological variables measured from 10:00 LT to 12:00 LT for each observation date were used for the model simulation to calculate snow albedo at 10:00 LT, 11:00 LT and 12:00 LT.

In this study, four kinds of snow albedo simulations were conducted based on four assumptions of snow impurity: 1. snow

albedo without any impurities present  (Alb-C), 2. snow albedo with the effect of MD only (Alb-D), 3. snow albedo with the effects of MD and BC only (Alb-DB), and 4. snow albedo with the effects of all impurities i.e. MD, BC and algae (Alb-DBA).

## 3 Results

### 3.1 Temporal changes in physical properties of the snow

Surface albedo of the snowpack at the study site gradually decreased with snow melting from late June to early August (Figs.

2a, b). The snowpack melted continuously from day 176 (June 25, 2014) to day 215 (August 3, 2014), as previously described by Onuma et al. (2018). For example, surface snow was fresh snow on day 168, and then it became granular snow on day 176 and remained so until day 215. Mean optically equivalent snow grain size (radius) of the surface was $0.3 \pm 0.1$ mm (mean $\pm$ SD) on day 168, $0.6 \pm 0.4$ mm on day 176, and then it varied between 0.7 and 0.9 mm until day 215. The properties of subsurface snow changed similarly. Relative snow surface level (0 cm on day 168) gradually decreased by 123 cm during the

study period (from day 168 to day 215). Snow albedo was 0.791 on day 168, and then it gradually decreased until day 209 (from 0.791 to 0.698). Finally, it decreased rapidly by 0.08 from day 209 to day 215.





## 3.2 Temporal changes in impurities in snow

The mass concentration of MD in both surface and subsurface snow gradually increased from mid-June to early August at the study site and reached the maximum in early August. The concentration in surface snow was $2.7 \times 10^{-1}$ mg L$^{-1}$ on day 176
(June 25, 2014) and gradually increased, with a slight temporary decrease on day 209, but increased again and finally reached $7.5 \pm 2.9 \times 10$ mg L$^{-1}$ on day 215 (Fig. 2c). A statistical test (one-way analysis of variance (one-way ANOVA)) demonstrated that the temporal change in the mass concentration of MD was significant ($F = 4.95$, $P = 0.03 < 0.05$). The concentration in the subsurface snow was generally lower than that in surface snow. It was $3.3 \times 10^{-1}$ mg L$^{-1}$ on day 176 and gradually increased to $1.4 \times 10$ mg L$^{-1}$ on day 215 (Fig. 2c).

In contrast to MD, the mass concentration of BC did not show seasonal trends either in surface or subsurface snow during the study period. The BC concentrations in surface and subsurface snow ranged from $5.4 \times 10^{-5}$ to $2.5 \times 10^{-2}$ mg L$^{-1}$ (mean: $9.5 \times 10^{-3}$ mg L$^{-1}$) and $1.2 \times 10^{-5}$ to $1.8 \times 10^{-2}$ mg L$^{-1}$ (mean: $3.5 \times 10^{-3}$ mg L$^{-1}$), respectively (Fig. 2d). The temporal change of BC was not statistically significant for either surface or subsurface snow (one-way ANOVA, surface: $F = 3.14$, $P = 0.11 > 0.05$; subsurface: $F = 8.89$, $P = 0.37 > 0.05$).

The mass concentrations of OC in surface snow gradually increased from mid-June to early August, and it was maximal in early August. The OC concentration in surface snow was $3.2 \times 10^{-2}$ mg L$^{-1}$ on day 168 and gradually increased to $3.4 \pm 0.3 \times 10^{-1}$ mg L$^{-1}$ on day 215, although the concentration decreased temporally on days 197 and 209 (Fig. 2e). This temporal change was significant in surface snow (one-way ANOVA, $F = 3.14$, $P = 9.8 \times 10^{-7} < 0.01$). Mass concentration of OC in the subsurface snow ranged from $3.8 \times 10^{-2}$ to $2.0 \times 10^{-1}$ mg L$^{-1}$ (mean: $8.4 \times 10^{-2}$ mg L$^{-1}$; Fig. 2e). There was no significant difference in
concentration from day 168 to day 215 (one-way ANOVA, $F = 8.89$, $P = 0.18 > 0.05$).

The concentration of OC was positively correlated to algal cell concentration of *Cd. nivalis*, as previously quantified by Onuma et al. (2018). The concentration of algal cells ranged from 0 to $4.9 \times 10^4$ cells L$^{-1}$ in the surface snow from day 168 to day 215 (Fig. 3). The relationship between algal cell and OC concentrations exhibited a significant positive linear correlation (r = 0.93, $P = 8.0 \times 10^{-4} < 0.05$). Based on the relationship, a regression line was obtained as follows:
$$C_{OC} = 5.3 \times 10^{-6} C_{Algae} + 0.0826 \tag{3}$$

where, $C_{Algae}$ is algal cell concentration (cells L$^{-1}$) in surface snow. This $C_{OC}$ was used as an input variable for simulation of PBSAM in this study.

## 3.3 Temporal changes in snow albedo simulated with PBSAM

Snow albedos simulated with the effect of only MD (Alb-D), with MD and BC (Alb-DB) and with MD, BC and snow algae
(Alb-DBA), gradually decreased from mid-June to early August, whereas the snow albedo calculated without the effect of any impurity (Alb-C) did not change significantly (Fig. 4). Physical properties of snow, snow impurities and meteorological conditions from 10:00 LT to 12:00 LT and used as input data for the simulation, are presented in supplementary material (Table S1-S5). Mean Alb-C ranged from 0.709 to 0.753 during the study period, but the change was not statistically significant.



Mean Alb-DBA was 0.755 on day 168 and gradually decreased to 0.687 until day 209, followed by a large decrease to 0.616

on day 215. The temporal changes of Alb-D and Alb-DB were similar to that of Alb-DBA. Coefficients of determination for the regression ($R^2$) between the calculated and observed albedo from day 168 to day 215 were 0.38 for Alb-C, 0.94 for Alb-D, 0.93 for Alb-DB and 0.93 for Alb-DBA. The root mean square errors (RMSEs) were 0.04 for Alb-C, 0.02 for Alb-D, 0.02 for Alb-DB and 0.02 for Alb-DBA. These results indicate that the three albedo simulations including snow impurities (Alb-D, Alb-DB and Alb-DBA) exhibited good performance in representing temporal changes of measured albedo.

**4 Discussions**

**4.1 Temporal changes of MD and BC on the snowpack**

Differences in temporal changes in mass concentrations of MD and BC suggest that they were transported to the snow surface through different processes. Aoki et al. (2014b) have reported temporal changes in MD and BC concentrations in surface snow located at an elevation of 1490 m a.s.l. in northwest Greenland (SIGMA-A site, 78°03' N, 67°38' W). During

their observations, from June 28 to July 12, 2012, the MD concentration in surface snow increased 349 times and reached 1.3 mg L$^{-1}$ while BC concentration increased only 5.4 times and reached $4.9 \times 10^{-3}$ mg L$^{-1}$. Their study suggested that the main factors explaining the increment were deposition from the atmosphere for MD and an enrichment following sublimation and evaporation of snow for BC. Geochemical analyses of MD on Glaciers in the Arctic region, including Qaanaaq Glacier, suggested that it is likely to be supplied mainly from local ground surfaces (e.g. moraine near the glacier) rather than more

distant areas (Nagatsuka et al., 2014; 2016, Tobo et al., 2019). The mass concentration of MD at the study site increased 57 times from $2.7 \times 10^{-1}$ mg L$^{-1}$ on day 176 (June 25, 2014) to 15.2 mg L$^{-1}$ on day 190 (July 9, 2014). The increase of MD concentration at the study site was likely due to exposure of the ground surface during the melting season. The mean BC concentration at the study site was $9.5 \times 10^{-3}$ mg L$^{-1}$, which is same order to those (from $0.9 \times 10^{-3}$ to $4.9 \times 10^{-3}$ mg L$^{-1}$) at SIGMA-A site, suggesting that BC at both sites was supplied from distant sources.

Temporal changes of calculated albedo suggest that MD is the main factor causing the reduction in albedo at the study site during the melting season. Alb-D and Alb-DB gradually decreased from mid-June to early August, whereas Alb-C did not change significantly during that period (Fig. 4). Thus, the reduction in albedo at the study site was due to the increase of snow impurities, rather than the changing of the snow grain size. The snow grain size did not change significantly after day 168. The differences of surface albedo between the Alb-C and Alb-D, and between the Alb-D and Alb-DB were 0.1 and almost 0 on

day 215, respectively, which were equivalent to the reduction in albedo caused by MD and BC, respectively. Although the MAC of BC used was larger than that of MD at any wavelength (Table 1), the effect of BC on albedo was smaller than the effect of MD. This was due to the greater concentration of MD on the snow surface.



## 4.2 Reproduction of temporal change in snow albedo using PBSAM, including the effects of snow algae

Temporal changes in algal cell concentration were positively correlated with that in the mass concentration of OC in surface

snow, suggesting that snow algae can be regarded as the main constitution of OC in snow (Fig. 3). The positive correlation between the observed algal cell and OC concentrations in surface snow suggests that OC in snow can be approximated using the algal cell concentration at the study site. Indeed, *Cd. nivalis* was the dominant species in snowpack at the study site throughout the summer season of 2014 (Onuma et al., 2018). However, significant amount of OC was detected in snow samples without algal cells, indicating that these snow samples contained non-algal organic matter, probably atmospheric OC aerosol.

The intercept of 0.0826 of equation (3) can be interpreted as to be contributed from the atmospheric OC aerosol other than algae. Although non-algal organic matter possibly affects albedo simulations using PBSAM, with MAC for snow algae, this effect was neglected in the present study because the concentration was much smaller than algal organic matter at the study site.

The MACs for snow algae in this study were likely to reproduce characteristics of light absorption caused by a snow algal

bloom. In order to calculate the effects of light absorption by snow algae on snow albedo, we assumed that four pigments (chlorophyll-a, chlorophyll-b, primary carotenoids, and secondary carotenoids) accounted for absorption in each cell. These pigments are major light abruption pigments for red snow algae (Cook et al., 2017a). Spectral variation of the imaginary part of refractive indices for wavelengths in this study, calculated from the four algal pigments, agreed with the spectral variation of that estimated from observed spectral reflectance of the red snow surface in the Qaanaaq Glacier (Aoki et al., 2013). The

dominant species in the snowpack of the glacier was *Cd. nivalis* (Onuma et al., 2018). These results suggest that the imaginary part of refractive indices in this study can reproduce the light absorption based on theoretical optical characteristics of *Cd. nivalis*.

Temporal change of snow albedo (Alb-DBA) on the snowpack at the study site was simulated with the PBSAM, including the effects of the three impurities (MD, BC, and snow algae), for the study period (from day 168 to 215). The result indicates

that the model that included the total effect of MD, BC, and snow algae was the best in reproducing temporal changes throughout the study period. The values of $R^2$ and RMSE between the observed albedo and modelled Alb-DBA from day 168 to 215 were 0.93 and 0.016, respectively. Furthermore, the Alb-DBA exhibited good performance in simulations with MD and BC only (Alb-DB). However, there was no significant difference in model performance among these simulations. This is probably due to the lower cell concentrations at the study site, which was $4.9 \times 10^4$ cells L$^{-1}$ on day 215, when compared with

those of typical red snow, which range from $3.2 \times 10^6$ to $2.3 \times 10^8$ cells L$^{-1}$ (Sutton et al., 1972; Thomas and Duval, 1995; Painter et al., 2001; Takeuchi, 2013; Lutz et al., 2014; Onuma et al., 2018). The spectral reflectance of the surface snow was consistent with this; it did not show the typical spectral absorption of the algal pigments (carotenoids and chlorophylls), which have absorption peaks in the wavebands of 400-600 nm and 670-680 nm (Painter et al., 2001; Takeuchi et al., 2006). The snow albedo simulation by Cook et al. (2017a) suggested that algal abundance of 10 μg$^{algae}$ / g$^{snow}$, which is equivalent to $5.9 \times 10^4$

cells L$^{-1}$ for a snow density of 600 kg m$^{-3}$, has little effect on the spectral absorption between 400 to 2200 nm.





### 4.3 Simulation of temporal changes in surface snow albedo using PBSAM and an algal growth model

Although our field observations ended on day 215, snow algae could grow further and increase their abundance until the end of the melting season. In order to infer temporal changes in snow albedo for the whole melting season, we calculated snow albedo using the PBSAM and a snow algal growth model proposed by Onuma et al. (2018). Temporal changes in algal abundance on surface snow of Qaanaaq Glacier can simply be expressed by a differential logistic growth equation. Microbial growth was therefore calculated as follows (Onuma et al., 2018):

$$X = \frac{K}{1+\frac{K-X_0}{X_0}e^{\mu(t_0-t)}} , t = d - d_f \tag{4}$$

where $X$ and $X_0$ are population densities of microbes at $t$ and $t_0$, respectively, and $\mu$ is the growth rate of microbes in $t^{-1}$. $K$ is the carrying capacity of algae in the snow surface and $t_0$ is the day of the first appearance of algae on the snow surface. $t$ represents the number of the days during which the snow surface temperature was above 0°C, because snow algae can grow only on the melting snow surface. Snow surface temperatures at the study site were obtained from the AWS data (Onuma et al., 2018). Parameters for the algae, including the initial cell concentration, algal growth rate and carrying capacity, were also the same as those in the previous study ($9.0 \times 10^{-1}$ cells L$^{-1}$, 0.39 day$^{-1}$ and $3.2 \times 10^6$ cells L$^{-1}$, respectively; Onuma et al., 2018). Because the snowpack at the site was unlikely to have disappeared after day 215, snow algae possibly grew on the surface snow beyond day 215. The calculation showed that algal cell concentration significantly increased from day 216 to 233 ($1.5 \times 10^5$ to $1.6 \times 10^6$ cells L$^{-1}$), and then remined constant until the end of the melting season (Fig. 5).

Simulation of snow albedo using PBSAM, coupled with the snow algal growth model, was conducted on the snowpack at the study site for entire melting season in 2014 (Fig. 5). The meteorological conditions, snow physical properties and inorganic impurities were assumed to be constant after the last observation on day 215 (Table S1-S5). Consequently, Alb-DBA was 0.616 on day 216 and 0.612 at the end of melting season (day 233). The effects of the snow algae (the difference between Alb-DB and Alb-DBA) were 0.001 and 0.004 on day 216 and 233, respectively, indicating that snow albedo was significantly decreased owing to the snow algae blooming late in the melting season (August).

### 4.4 Possible albedo reduction in the presence of typical red snow algal blooming

We validated the albedo reduction for high algal abundance using the snow albedos of typical red snow surface reported by Painter et al. (2001) and Lutz et al. (2014). The algal cell concentrations reported in their studies were used as input variables in surface (0–2 cm) and subsurface (2–10 cm) snow (Painter et al., 2001: $2.1 \times 10^7$ cells L$^{-1}$; Lutz et al., 2014: $1.8 \times 10^6$ cells L$^{-1}$ at site MIT-17). These algal cell concentrations were converted into $C_{OC}$ using equation (3). Our observational data on day 215 (meteorological, snow physical and impurity conditions) were used as other input data of these simulations. The simulation using the cell concentration reported by Painter et al. (2001) demonstrated that the difference between Alb-DB and Alb-DBA was 0.062, which is equivalent to the albedo reduction by snow algae, and in agreement with the algal albedo reduction (0.07) observed by Painter et al. (2001). This reduction in albedo was also close to the result of another simulation with the bio-albedo model proposed by Cook et al. (2017a, algal albedo reduction = 0.07). Thus, both our PBSAM and the bio-albedo





model can consistently reproduce the reduction in albedo based on the optical properties of *Cd. nivalis*. In contrast, the simulation using the cell concentration reported by Lutz et al. (2014) produced an albedo reduction by snow algae of 0.008,

which was lower than that observed by them (0.09) and calculated with the bio-albedo model (0.09). This is probably owing to different algal pigments in the ice surfaces. Lutz et al. (2014) reported that ice algae (filamentous cells: $6.1 \times 10^6$ cells L$^{-1}$) were found in addition to snow algae (spherical cells: $1.8 \times 10^6$ cells L$^{-1}$) in the samples collected at their study site (MIT-17). The pigments of ice algae have greater light absorption than that of *Cd. nivalis* (Dial et al., 2018). In the albedo simulation with the bio-albedo model, measured pigment compositions (total chlorophyll, primary and secondary carotenoids) were used

as model parameters while our simulation only used MAC for snow algae (*Cd. nivalis*). Although further study is necessary to validate the effect of algae on snow albedo, our results suggest that this albedo model has a potential to reproduce albedo reduction by snow and ice algae.

**4.5 Potential for albedo reduction caused by red algal blooming**

Using the PBSAM, we conducted sensitivity analyses to quantify the reduction in albedo with different concentrations of

snow algae. Figure 6 shows the albedo reduction by snow algae (Alb-DB minus Alb-DBA) as a function of algal cell concentration for various snow grain sizes or MD concentrations on the surface snow. Algal cell concentrations ranged from $4.9 \times 10^4$ to $2.3 \times 10^8$ cells L$^{-1}$, which cover the range of cell concentrations for typical red algal snow reported previously (Sutton et al., 1972; Thomas and Duval, 1995; Painter et al., 2001; Takeuchi, 2013; Lutz et al., 2014; Onuma et al., 2018). Various snow grain sizes and MD concentrations were used in the simulation (0.3–1.5 mm and 0–150 mg L$^{-1}$, respectively),

which were based on observations on day 215 in this study (0.87 mm and 75 mg L$^{-1}$, respectively). Observational data on day 215 (meteorological, snow physical and snow impurities conditions) were used as other input data for this simulation. The simulation demonstrated that the albedo reduction by snow algae ranged from 0 to 0.207 (algal cell concentration: $4.9 \times 10^4$ to $2.3 \times 10^8$ cells L$^{-1}$) with an MD concentration of 75 mg L$^{-1}$, consistent with the algal albedo reduction estimated for red algal abundances observed on various arctic glaciers (a maximum reduction of 0.2; Lutz et al., 2016). Thus, the simulation with our

albedo model was comparable to the visible red algal blooming.

**4.6 Light absorption of algal cells in different snow grain sizes and inorganic impurity concentrations in snow**

Sensitivity analyses with PBSAM using different snow algal cell concentrations and grain sizes suggested that increased snow grain size in a snow layer can enhance light absorption by snow algal cells arising from deeper penetration of incident radiation through snowpack. The difference between Alb-DB and Alb-DBA was larger when snow grain size was large (1.2

mm, albedo reduction: 0–0.22; Fig. 6a). Conversely, the difference was smaller when snow grain size was smaller (0.6 mm, albedo reduction: 0–0.19). Aoki et al. (2011) suggested that light penetration depth in snow composed of coarse grains is deeper, enhancing light absorption by inorganic impurities due to increased scattering of light. Therefore, increased snow grain size possibly enhances albedo reduction by snow algae. Red snow algal blooming could accelerate the increase of snow grain



size because of the increase in penetration of incoming radiation within the snowpack, leading to further reduction in snow albedo.

Increases of MD and BC concentrations in a snowpack possibly weaken the scattering of light in the snowpack and reduce the amount of light absorbed by snow algae. The difference between Alb-DB and Alb-DBA was smaller when MD concentrations were lager (100 mg $L^{-1}$, albedo reduction: 0–0.20; Fig. 6b). Conversely, the difference was larger when MD concentrations were reduced (50 mg $L^{-1}$, albedo reduction: 0–0.22). These results suggest that algal cells absorb more light
when MD concentrations were smaller, compared with a higher MD concentration. The albedos calculated with different BC concentrations also confirmed this result. The higher concentrations of MD or BC may decrease the intensity of light scattered in snow layers, thereby resulting in reduced light absorption by algae. There is limited information about the effect of MD and BC on algal light absorption in snowpacks, but a recent study suggested that the interaction between algal cells and other impurities in snow should be investigated (Cook et al., 2017a). Although further study is necessary to investigate in situ
interactions among snow algae and inorganic impurities in snowpack, our simulation suggests that increased concentrations of inorganic snow impurities weaken algal light absorption due to a reduction of the intensity of scattered light in snow.

### 4.7 Radiative forcing of algal cells

To quantitatively assess the effects of snow algal blooming on the net shortwave radiation, we calculated the radiative forcing from the observed downward shortwave radiation multiplied by the difference of Alb-DB and Alb-DBA, following
the method of Niwano et al. (2012). Meteorological conditions measured from 10:00 LT to 12:00 LT on day 215 were used to calculate the radiative forcing. The calculations demonstrated that the radiative forcings ranged from 0 to 0.1 W $m^{-2}$ (mean: 0.1 W $m^{-2}$) and 19.0 to 67.3 W $m^{-2}$ (mean: 39.7 W $m^{-2}$) when cell concentrations were $4.9 \times 10^4$ and $2.3 \times 10^8$ cells $L^{-1}$, respectively. The difference of Alb-DB and Alb-C was used to calculate the radiative forcing of total MD and BC, and on day 215 at the study site the forcing ranged from 8.4 to 33.9 W $m^{-2}$ (mean: 19.4 W $m^{-2}$). Cell concentrations of *Cd. nivalis* in visibly
red snow surfaces ranged from $1.0 \times 10^6$ to $5.0 \times 10^7$ cells $L^{-1}$ in Greenland (Lutz et al., 2014; Lutz et al., 2016; Onuma et al., 2018). The radiative forcings calculated with these cell concentrations were equivalent to the range from 0.2 to 8.3 W $m^{-2}$. Our calculations suggest that prominent red algal blooming ($5.0 \times 10^7$~ cells $L^{-1}$, equivalent to 300~ mg $L^{-1}$) has the potential to increase radiative forcing equal to that caused by total MD (75 mg $L^{-1}$) and BC ($3.7 \times 10^{-3}$ mg $L^{-1}$), although further field observation and model validation in various snowfields are needed to discuss the potential for studying albedo reduction arising
from red algal blooming.

### 4.8 Temporal changes in snow albedo reduction caused by red algal growth under warming condition

The scenario simulations of PBSAM, coupled with the snow algal growth model, suggested that albedo reduction by snow algae reached a maximum of 0.04, equivalent to a radiative forcing of 7.5 W $m^{-2}$, when surface snow temperature was increased by 1.5°C in August, 2014, at the study site. Monthly mean surface air temperature at the study site, from 2012 to 2017, which
was measured with the SIGMA-B AWS, ranged from -2.9 to 0.2°C in August (2014 season: -1.2°C). Because snow algae can



grow continuously during snow melting (Onuma et al., 2016; 2018), the abundance of snow algae at the study site could differ each year. In this study, we simulated temporal changes in snow albedo during the late summer season using PBSAM coupled with an algal growth model, while assuming various surface snow temperatures to estimate the impact of red snow algal growth on snow albedo under global warming (Fig. 7). The simulation was conducted for 30 days, starting after day 215. The temporal

changes in surface snow temperature under different assumptions were used as input variables for the algal growth simulations (surface snow temperature of plus or minus 1.5°C). The initial cell concentration, algal growth rate and carrying capacity, were $4.9 \times 10^4$ cells L$^{-1}$, 0.39 day$^{-1}$ and $2.3 \times 10^8$ cells L$^{-1}$, respectively. Observational data on day 215 were used as the other input variables. Our simulations suggested that snow algae can exhibit additional growth in warmer conditions, resulting in a larger reduction in albedo, equivalent to larger radiative forcing (Fig. 7). In particular, simulations with surface snow temperature of

plus 1.0–1.5°C demonstrated that the reduction in albedo and radiative forcing significantly increased for 30 days (gray shading in Fig, 7). Although there is little information pertaining to red algal blooming on surface snow in Greenland, satellite observations have detected red algal blooming on surface snow caused by growth of *Cd. nivalis* in Southeast Greenland (Hisakawa et al., 2015). Ganey et al. (2017) suggested that the red snow area detected by Landsat 8 extended over about 700 km$^2$ on an Alaskan icefield, and the red snow was responsible for 17% of the total snowmelt there. Further study is necessary

to simulate snow albedo with the inclusion of the effect of red algal growth over the Greenland Ice Sheet. Future climate warming in Greenland may expand the area of red snow in the near future, leading to accelerated loss of mass balance of the Greenland Ice Sheet.

## 5 Conclusions

    Temporal changes in snow albedo of Qaanaaq Glacier in northwest Greenland were calculated with a physical snow albedo

model that incorporated the effect of three snow impurities (MD, BC and snow algae). A PBSAM (Aoki et al., 2011) can calculate snow albedo using meteorological conditions, snow physical properties and snow inorganic impurities. To quantify the effect of snow algal blooming on snow albedo, we calculated a light absorption coefficient for red snow algae, based on geometry and the pigment composition of red algae (*Chlamydomonas nivalis*) and introduced this coefficient into PBSAM. In addition, we simulated a temporal change in snow albedo using this PBSAM coupled with a simple numerical model for snow

algal growth (Onuma et al., 2018). The calculated albedo agreed with the observed albedo during the algal growing period, from late June to early August, although the algal cell concentration did not reach the level of typical red snow blooming during the observation period. We also calculated the snow albedo of the typical snow algal blooming surface previously reported and demonstrated that it agreed with the observed snow albedo. Our simulation suggests that typical red snow algal blooming has a potential to reduce snow albedo by 0.21, equivalent to a radiative forcing of 40 W m$^{-2}$. Finally, we conducted

scenario simulations (surface snow temperature of plus or minus 1.5°C) in order to estimate a possible albedo reduction by snow algae in the near future. The albedo reduction by snow algae only equaled 0.04 (radiative forcing: 7.5 W m$^{-2}$) during a warmer ablation season (surface snow temperature of +1.5°C) in northwest Greenland, suggesting that climate warming of the

near future of Greenland Ice Sheet may expand the area of red snow and further accelerate a loss of the mass balance. Our model can simulate surface albedo in broadband wavelength (300-3000 nm) range, including the effects of both organic and

inorganic impurities, and can independency estimate the reductions in albedo arising from each impurity (MD, BC and snow algae). Inter-comparison with other albedo models (e.g. the bio-albedo model proposed by Cook et al., 2017a) would be useful to develop the albedo model and to further understand the process of albedo reduction arising from microbial activities on snow and ice. Although further study is necessary to understand dynamics of organic and inorganic impurities in the snowpack, the physical model of snow albedo coupled with algal growth model has the potential to provide a mechanistic understanding

of temporal changes of snow albedo over the Greenland Ice Sheet by incorporating microbial activity on the snow and ice. In future, coupling a reginal climate model NHM-SMAP (Niwano et al., 2018), which use PBSAM as snow albedo scheme, and the algal growth model, will enable us to estimate the effect of algal blooming on the melting of snow.

**Data availability**

All of the observation and model input and output data presented in this study are available upon request to the corresponding

author (Yukihiko Onuma, onuma@iis.u-tokyo.ac.jp).

**Author contributions**

YO and NT designed the study and wrote the paper. YO and TA established light absorption coefficient for red snow algae and simulated snow albedo with PBSAM. YO, ST and NN collected snow samples and observed snow physical properties. YO and ST analyzed the collected data. MN and TA prepared the SIGMA AWS data and provided technical support.

**Competing interests**

The authors declare that they have no conflict of interest.

**Acknowledgements**

We would thank to the filed campaign members of the SIGMA (Snow Impurity and Glacial Microbe effects on abrupt warming in the Arctic) Project and GRENE (the Green Network of Excellence) Arctic Climate Change Research Project in

Greenland in 2014. This study was supported in part by Grant-in-Aids (23221004, 26247078, 26241020, 16H01772, 19H01143) and Arctic Challenge for Sustainability (ArCS) project.



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





**Table 1: Mass absorption coefficients (MACs) of snow impurities parameterized in the PBSAM (Aoki et al., 2011). The MAC for red snow algae was assumed in our study.**

| Wavelength (nm) | Mineral dust ($m^2\ g^{-1}$) | Black carbon ($m^2\ g^{-1}$) | Red snow algae ($m^2\ g^{-1}$) |
|---|---|---|---|
| 200–400 | 1.5E-01 | 1.5E+01 | 2.0E-02 |
| 400–475 | 9.0E-02 | 1.3E+01 | 5.8E-02 |
| 475–550 | 6.2E-02 | 1.2E+01 | 5.8E-02 |
| 550–625 | 4.1E-02 | 1.0E+01 | 4.8E-02 |
| 625–700 | 3.5E-02 | 9.2E+00 | 4.0E-02 |
| 700–950 | 2.8E-02 | 7.4E+00 | 1.6E-03 |
| 950–1125 | 2.3E-02 | 5.4E+00 | 4.8E-05 |
| 1125–1400 | 2.3E-02 | 4.3E+00 | 2.2E-04 |
| 1400–1950 | 2.2E-02 | 3.0E+00 | 2.0E-03 |
| 1950–3000 | 3.4E-02 | 2.1E+00 | 7.5E-03 |

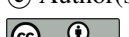




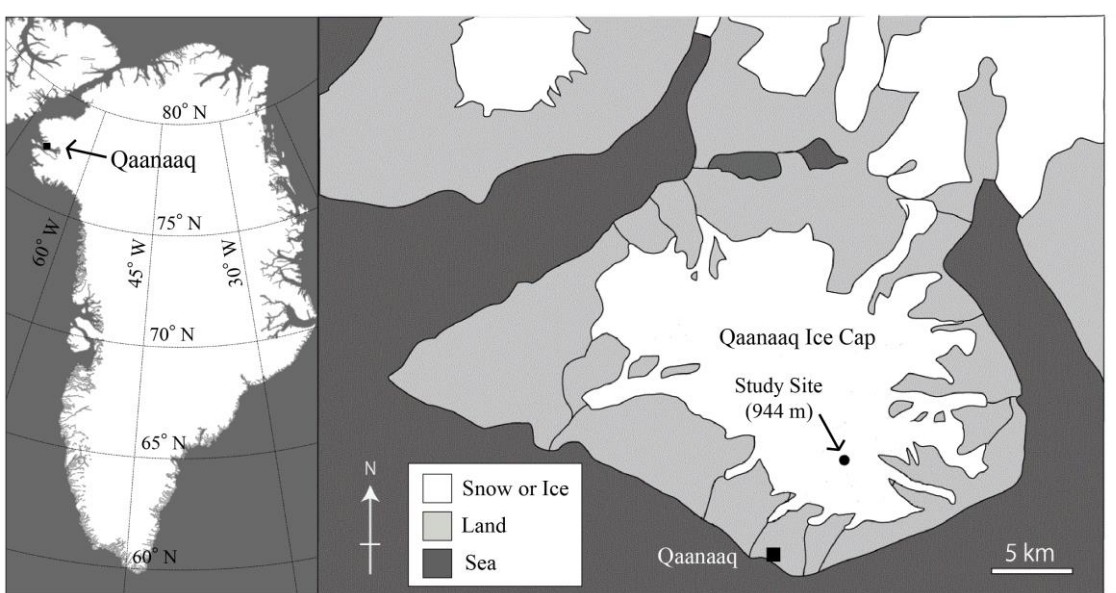

**Figure 1: Map of Greenland (left) and Qaanaaq Ice Cap in northwest Greenland (right). The figure to the right shows the sampling sites on the glacier.**





**Figure 2: Temporal changes in observed snow physical properties on surface and subsurface snow at Site B. (a) snow albedo, (b) snow grain size (radius), (c) mass concentration of MD, (d) mass concentration of BC, (e) mass concentration of OC. Snow albedo was calculated from the ratio of upward and downward shortwave radiation at AWS. Error bars = standard deviation.**


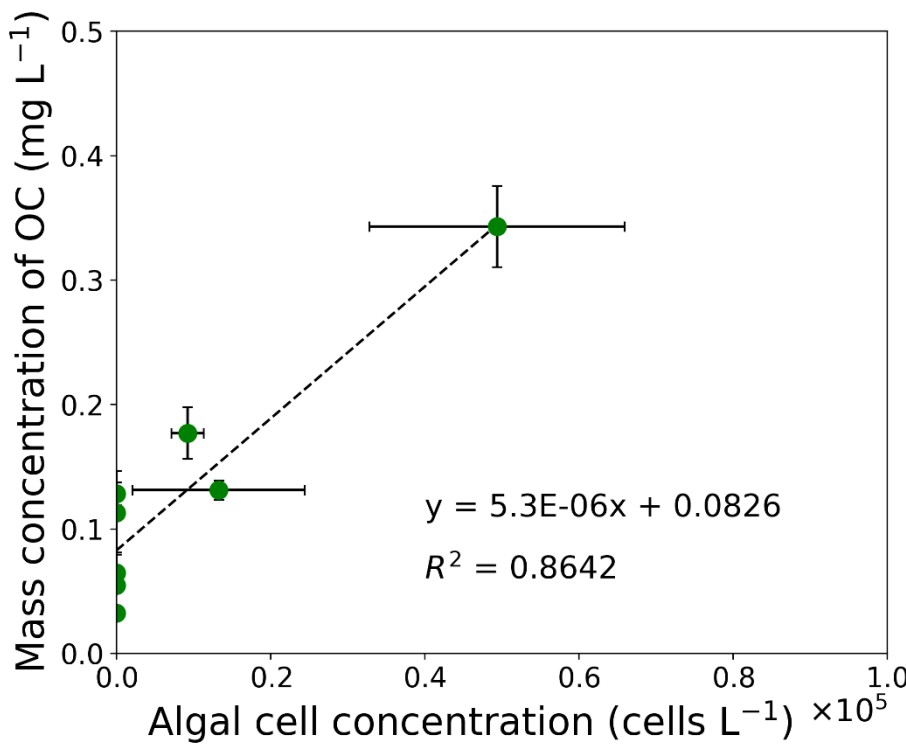

**Figure 3: Correlation chart between mass concentration of OC and algal cell concentration from filed measurements at the study site. The correlation coefficient is 0.93 (P = 8.0 × 10⁻⁴ < 0.05). Error bars = standard deviation.**




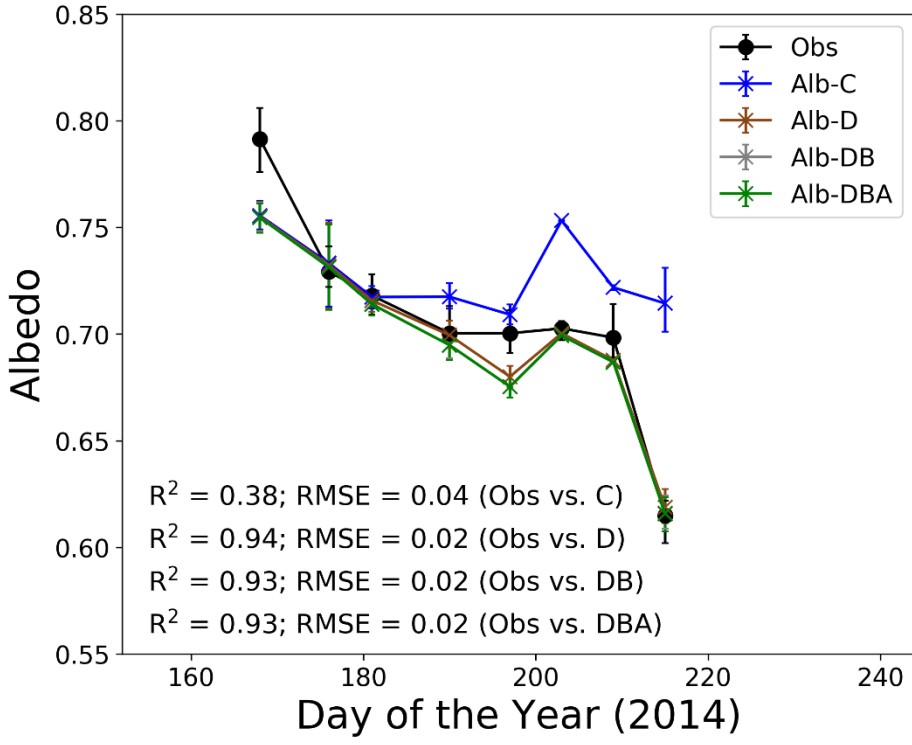


**Figure 4: Temporal changes in observed and calculated snow albedo at the study site. Solid symbols indicate observed snow albedo. Cross symbols indicate four simulations of snow albedo based on the assumption of snow impurity inclusions. Error bars indicate the albedo range simulated using the meteorological conditions at 10:00 LT, 11:00 LT, and 12:00 LT.**






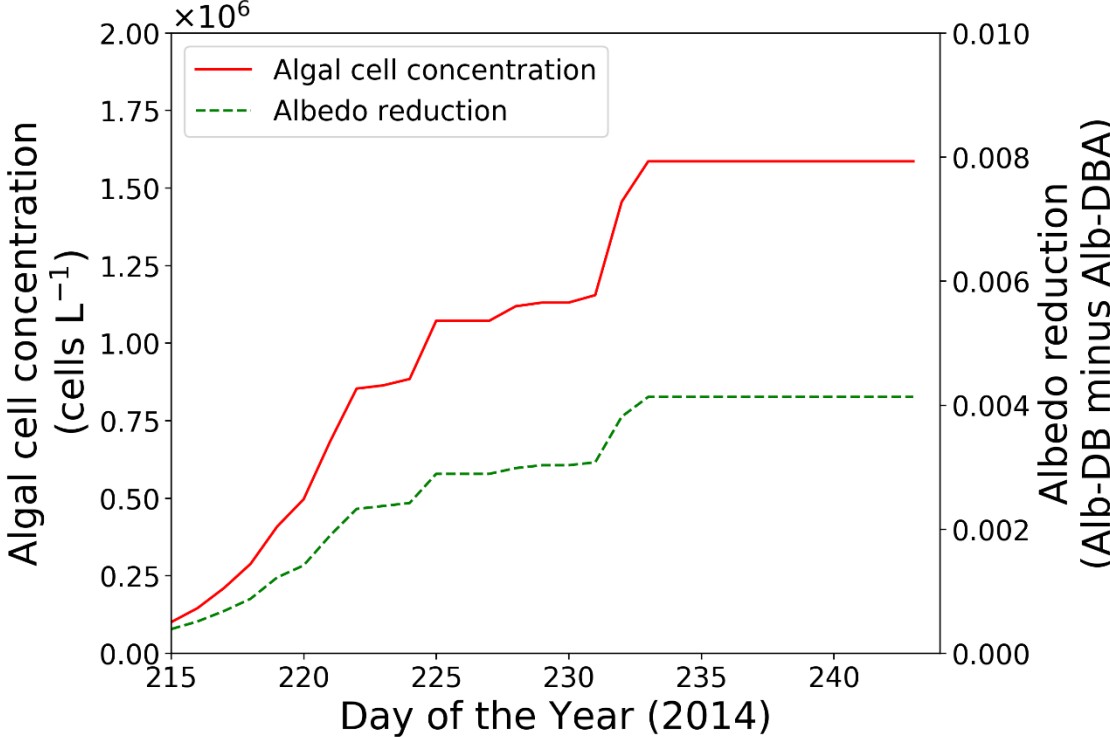

**Figure 5: Temporal changes in algal growth and albedo reduction from day 215 to 243 at the study site; algal cell concentration (red solid line) and albedo reduction (green dotted line).**






**Figure 6:** The relationship between reduction in snow albedo and algal cell concentration. Albedo reductions were calculated from the difference in Alb-DB and Alb-DBA simulated with an assumed algal cell density and (a) snow grain size, or (b) MD concentrations.
Grey shades indicate the range of the albedo reduction simulated with snow grain size ranging from 0.3 to 1.5 mm or MD concentration from 0 to 150 mg L$^{-1}$, respectively.





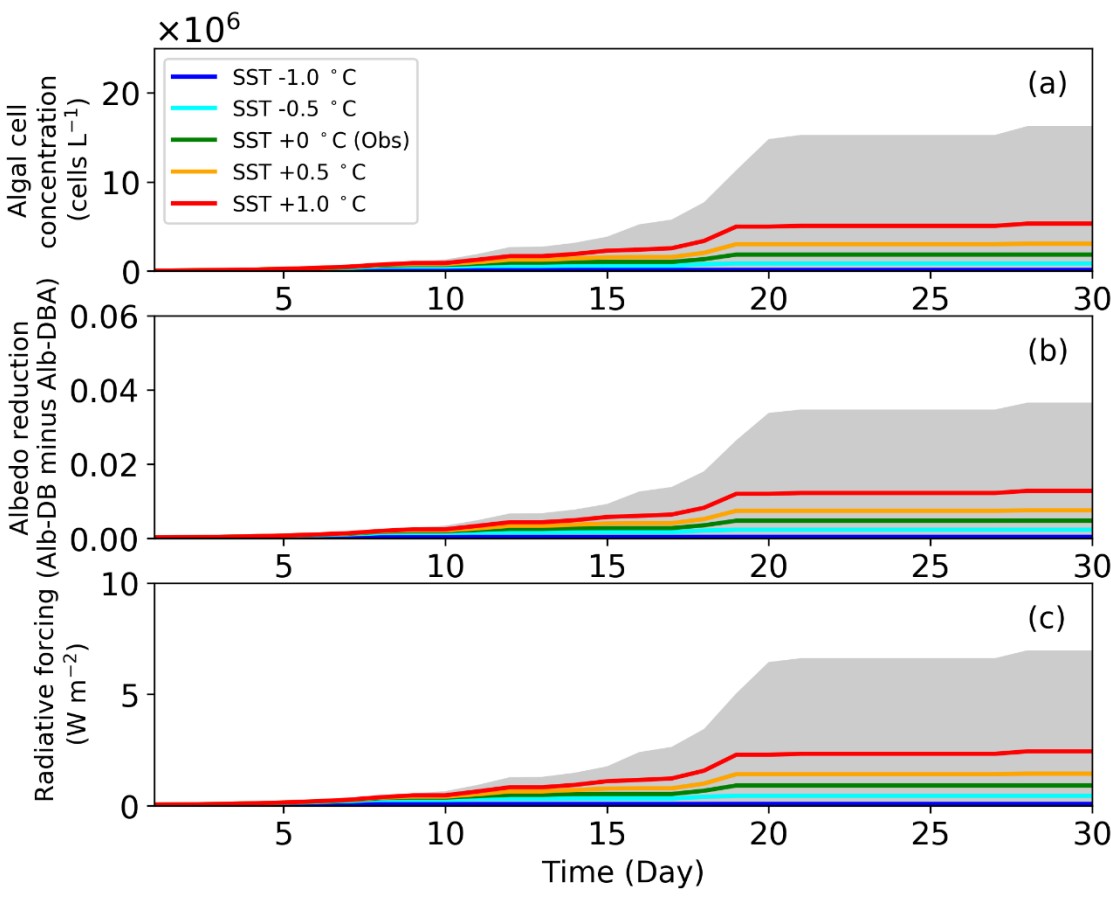

**Figure 7: Temporal changes in (a) algal growth, (b) albedo reduction by snow algae, and (c) radiative forcing by snow algae for 30 days under various surface snow temperature (SST) conditions at the study site. Grey shading indicates the range of each result simulated with surface snow temperature of plus or minus 1.5°C.**