# Peer review of "Physically-based model of the contribution of red snow algal cells to temporal changes in albedo in northwest Greenland"

_The Cryosphere, 2019_

## Referee Comment (RC1) · Daniel Remias (Referee) · 27 Dec 2019

This study links field observations of arctic snow packs at Greenland with phenomena influencing the surface albedo and thus alterations of melting rates with a mathematical model proposed by the authors. The model is intended to explain the role, respectively to forecast the consequences of different surface concentrations mineral dust, black carbon and organic impurities (mainly snow algae) on the seasonal melting rates. The questions of this work are of very high ecological relevance, taking global warming into account, and in this special case the fate of the Greenland Ice Sheet. Several im-

portant studies recently dealt with albedo changes of bare glacier ice, however partly neglecting the role of decreased albedo of snow on these glaciers, which will lead to earlier exposure of the glacier surfaces in the ablation zone during the melting season. The proposed model is feasible and a first good step for performing simulation of scenarios. As for every model, improvements will likely follow to make it more robust, and satellite data generally needs more accurate supplementation of field data acquired in situ. In general, false positive results have to be excluded. In the case of this work, one distinct glacier has been sampled during one season. My review will focus on biological aspects; I am not a specialist for albedo or mathematical simulations. The main issue of this manuscript is that the authors will have to consider the biology of algae causing red snow worldwide in more detail, and as a consequence, the model of snow algae "growth on snow surfaces" (which does practically not take place) has to be modified. Regarding terminology, "red algae" and "bloomings" should be avoided throughout the manuscript, and correctly "red snow algae" or better "blooms of red snow" should be used. Consequently, the title could be altered to i.e. "Temporal changes in snow albedo, including the possible effect of algae causing red snow, simulated with a physically based model" The globally most common algal species causing the well-known red snow phenomenon, also at Greenland, has been described recently as Sanguina nivaloides, and consequently "Chlamydomonas nivalis" should be avoided from now on. The according reference to be incorporated: Procházková et al. 2019 (https://doi.org/10.1093/femsec/fiz064). Likewise, the newest, updated general review for snow algae can be included: Hoham & Remias (https://doi.org/10.1111/jpy.12952; succeeding Hoham & Duval 2001). In these references, the authors can learn that the spherical red cysts which are abundant on snow surfaces, do not cleave at the surface during the melting season. Cell division takes place only early in the season when the population has not yet reached the still white surface, but the bottom snow is already water logged after the end of winter. This has a significant consequence for the proposed "snow algae growth factor": it does not exist! But from where comes the evident increase of seasonal cell concentration of such blooms? The only explanation

in my mind is, given that the algae population stays about the same during snowmelt (except that cell diameter will increase to a certain extend), that the volume of snow decreases due to melting (and partly sublimation). Thus, the cell numbers per snow volume increase only passively in the case of red cysts. For creating a model regarding the albedo issues caused by snow algae, the theory can be kept but renamed to i.e. "accumulation model". In this manuscript, the "typical" cell concentration per red snow volume is mentioned several times. But what is typical? First of all, it depends on the location of the ecosystem. Coastal snow fields are usually affected by local nutrient input and may have striking blood red coloration, whereas oligotrophic alpine and polar snow has much lower abundances, causing a more pink snow. Moreover, the concentration of the organic impurities on the snow surface can also be influenced by meteorological events like precipitation, and the authors should discuss this aspect. Melting surface snow has temperatures about 0.5 to 1 C in all studies I know. Therefore, why were calculations performed simulating an elevated snow temperature of +1.5C? This should be physically impossible, and thus I suggest removing this hypothetical data from the manuscript, and likewise change fig. 7.

Detail comments: Line 16, use Sanguina nivaloides instead of Chlamydomonas nivalis throughout Line 27: Why should this albedo model only be valid for snow covers in Greenland? Could it be applied elsewhere? Line 41: visible is from 400 to 700 nm Line 47: "of" instead "in" the Greenland Ice Sheet Line 50: insert "algae" to the list in brackets Line 52: insert "in" after "might be present" Line 57: Start sentence with "The ..." Line 67: reference Yallop et al. 2012 is about glacier algae, not about snow algae Line 68: delete "and ice" for snow algae. Line 70: give reference Hoham & Remias instead of Hoham & Duval Lnie 71: use Sanguina nivaloides instead of Chlamydomonas nivalis and include reference Procházková et al 2019 Line 74: "algal blooms have" instead of "algal blooming has" Line 78: "cellular pigment composition", delete "in cell" Line 84 & 93: reformulate as explained earlier: the algal abundance on surface is not due to growth but due to concentration. E.g. "accumulation model" Line 191: The mean cell radius was 11.4 $\mu$m – based on algae from the study site? Please state. Elsewhere in

the manuscript, a different cell size number based on an older study of Onuma et al. was given. Line 237: the decrease of OC on days 197 and 209 should be discussed, and please consider meteorological events like precipitation as a causer? Lines 240 – 245: The correlation between OC concentration and snow algae abundance appears to be obvious. Still, despite high significance, other organisms can also contribute to OC like bacteria, and in this habitat most likely, yeast-like fungi. This should be at least briefly discussed (e.g. line 285: the formula is not a proof that snow algae are the main constitute of OC in snow – though this is apparent for this study). Since fungal blooms depend on the snow algae, there could be a stable correlation as well. Line 285: "constituent" instead of "constitution" Line 289: please explain/hypothesise how algae-free snow contains significant amount of OC at a remote place of Northwestern Greenland. Long-distance deposition? Line 296: unneeded repetition of the considered pigments Line 297: "absorption", not "abruption" Line 299: Sentence "The dominant species was . . ." is a repetition of results. Line 309: the typical cell concentration of the study site was 4.9 x 10 e4 cells per litre. To my experience, the lower threshold to see snow discolorations by one's eye is about 5 x 10 e6. Thus my question, did you see red snow visually at the study site? This should be mentioned in the results. Line 310: delete non-existing reference "Sutton et al. 1972". There exists only a PhD of Mrs. Sutton from that year, a fine work though which was never published as a paper. There are many good papers for the concentration of Sanguina nivaloides in red snow. Chapter 4.3 (lines 316 – 337): I have serious problems with this part. It should either be rewritten or omitted. There are wrong biological assumptions about growth of red snow algal cysts (as stated in the beginning), which simply does not take place on the snow surface. The bloom that you can see is a (physiologically active) resting stage. Chapter 4.4: A "typical" red snow bloom should be defined more clearly. Only two references for average cell numbers are given. Furthermore, Painter et al 2001 is inappropriate in this context, because they clearly state that they did not any field measurements. Moreover, the number taken from Lutz et al 2014 from location MIT-17 is not the average value for red cysts in that study, but a rather low value for a bloom. For being repre-

sentative, I strongly recommend to include certain further studies of "Chlamydomonas nivalis" and then recalculate the albedo reduction of a "typical" red bloom. Line 353: Start sentence "The phenolic pigments of glacier algae have a broader bandwidth of spectral absorption than the carotenoids of S. nivaloides". Instead of citing Dial et al. 2018, which is rather a deductive/theoretical work, a reference showing real spectral data is more appropriate (e.g. Fig. 13.6 in Remias 2012, Springer Vienna) Line 363: Again, Sutton 1972 is not a paper and Painter et al 2001 give no numbers to cite for. Line 388: replace "but" with "and" Chapter 4.8: Needs to be updated. Please think over the "snow algae growth model" and the sense of using hypothetical high snow temperatures for any calculations. Growth of snow algae takes place in deep layers of snow with very constant conditions around 0.5°C. The air temperature plays generally no big role since the snow pack starts to be water logged from the bottom, and this occurs earlier than air temperatures raise above zero. Line 586: Reference Painter et al 2001 incomplete, authors missing. "Thomas" instead of Thimas".

Legend to fig 1: "sampling site", not "sites" Legend to fig. 5: say "Temporal changes in algal cell concentration and . . ." Figure 7: This is a bit vague considering the theory of cell growth and snow temperatures of +1.5°C

Please also note the supplement to this comment:
https://www.the-cryosphere-discuss.net/tc-2019-263/tc-2019-263-RC1-supplement.pdf

---

## Referee Comment (RC2) · Marian Yallop (Referee) · 3 Mar 2020

The role played by snow algae in contributing to albedo is currently of relevance and likely to be of interest to the larger cohort of scientists working in this field. This paper describes seasonal changes in the 'cyst stage' of a species they have identified as mostly, 'Chlamydonomas nivalis' type cells, documenting changes in numbers with time and modelling the contribution these cells make to a physically based snow albedo model. This species has now been renamed as Sanguina nivaloides (Procházková et al. 2019. The measurements have been made on a snowpack in northwest Greenland

during the ablation period of 2014. Samples were also collected to quantify mineral particles and organic and black carbon content. However, an assumption is made that the cells are actively growing on the ice, i.e. that cell division leads to an increase in numbers through time. However, the cysts do not divide on the surface of the snow, but will, once buried, potentially provide the inoculum for the following year should suitable conditions arise. In the spring or early summer, once melt water forms, the flagellated algal cells will swim up to the snow surface to form a bloom. On the surface, they lose their flagella and formation of the red pigments occur. Members of the so-called water-melon snow are likely more speciose than was once thought. The development of blooms on the surface can show considerable spatial and temporal heterogeneity, at any one point in time, making it difficult to reliably quantify their distribution. This patchiness may be due to species-specific pigment differences as well as change in their relative abundance, and these factors needs to be carefully considered for inclusion in any model where the aim is to quantify their contribution to albedo. However, I do not have expertise in the modelling sections and my comments refer to the biological sections of the manuscript.

It is recommended that consideration be given to rewording the title for two reasons: i) the named algae are green algae, members of the Chlorophyta or green algae, and readers may be confused into thinking that this paper is about red algae (Rhodophyceae); ii) these particular algae are not considered to grow (as in cell division) once on the surface; they rapidly form cysts, though the cysts themselves may potentially show size changes through time, and they can still be photosynthesising but at very low rates i.e. they can still be metabolically active if the conditions are right. But, it is these cysts that will eventually act as the inoculum for the next year hence their persistence on the surface is fundamental for species survival from year to year. They also form an important food source for a number of grazers. Through the text, including the abstract, it is important that the term red algae is revised accordingly. It needs to be emphasised that the cells, on reaching the surface would turn into cysts and that an increase in the concentration of cell numbers on the surface is as a result

of cell accumulation as the snow melts and algae concentration as they get behind. For this reason, all reference to algal growth must be removed. That said, it is possibly that some other microbes may be found growing on the snow but the particular species mentioned here would not. The researchers may find these additional papers of use. Fogg (1967) Phil. Trans. Soc , 252, 279-287. Fjerdingstad et al., 1974, Arch. Hydrobiol. 73, 70-83 as well as the newer review by Hoham & Remias, Journal of Phycology, 2020. Line 27, clarify what is meant by proper estimates? Suggest revise wording; Line 33, add Yallop et al., 2012 to the references reporting changes in surface albedo with ice melt; kline 41 – check all references made to ranges for visible light as they vary through the text; Line 45 – What is meant by ppbw (provide full definition); Line 46 Suggested reordering of this sentence: ….. was reported to be lower, by 0.7%, than that by BC…; Line 49 – change absorb to absorbs; line 51 – after present add an on; Line 57 Revise to read A physically based snow …….; Line 83 – and elsewhere – revise references to 'growth' of cells on the surface and references to growth model where the 'growth' is likely resulting from an accumulation of cells in a defined area rather than active division of cells. line 65, Yallop et al 2012 discussed potential ice algal albedo impacts not snow; line 96, Field not filed; Line 103 – through more recent molecular work it is likely that any surface blooms may contain a number of different species that have very similar morphologies and it may be better to use Chlamydomonas spp., to infer that. There is also an indication that there were some cells that were not spherical red cells. Can any more information be provided regarding the identity of these cells?; line 116. It would be useful to add more information about the density samples (make, model); line 120 . It is recommended that more information is provided in the text to provide details about the spatial sampling protocol. Line 135 – change bag to bags; Line 145 – after USA), change the 'in' to an 'on';Line 145 – were the samples preserved or rather maintained. Using the term may imply some preservative was added. Line 192 cell sizes were measured. Do these sizes account for any shrinkage as a result of the preservative used?; Line 194 and 195– remove the 's' from compositions; Line 195 – revise sentence as meaning is not

clear; Line 245 – Why is there a – sign in the equation for cell numbers?; line 261: Relevant to this comments is the potential for aggregation of material on the snow. The cells can be sticky and aggregate to form larger clumps, together with mineral particles and other associated matter, including bacteria. Aggregation may affect their motility. line 285 – change the word 'constitution' for 'constituent'; Line 268 – change 'amount' to 'amounts'; Line 309, here and elsewhere in text, if the cell numbers being reported are averages, can the SD or SE of cell number be added ;351 more literature and information could be added to support this statement that there may be different pigments in ice surfaces. The authors are referred to the new paper by Williamson et al. 2020 (PNAS , www.pnas.org/cgi/doi/10.1073/pnas.1918412117), for further views on the role of pigments. Importantly, whilst it is possible that snow algae may be found in ice environments, some of the major players on the ice sheet e.g. Ancylonema may not grow on the snow. Further, the latter species is actively growing and not in a resting stage hence it might be expected that their pigments would be very different. Line 410 onwards – references of growth of snow algae need to be removed here, though some snow algae may have vegetative stages in the snow.

Many references are made in relation to the growth of the snow algae on the surface, as the snow is melting. It would be very useful to add in more detail about the life cycle of the collective group of snow algae, detailing the light triggers that promote the of the biflagellate stage to the ice surface and their modification to form the resulting cyst stage, with loss of the flagellae. The transition period from one form to another, and the time period over which this may happen, is critical. Fogg et al. (1974) report that the increase in cell numbers in snow is sometimes as a result of cell concentration due to ablation (sublimation) which leaves the algae behind. Previous researchers report on active photosynthesis in these surface cells, though possibly activity would cease once the surface temperatures become too high. The review of Hoham and Remias, (2020) would also be useful here.

---

## Author Comment (AC1) · 13 Apr 2020

**Reply to Dr. Daniel Remias (Reviewer#1)**

April 3, 2020
Dr. Yukihiko Onuma
Institute of Industrial Science, University of Tokyo
E-mail: onuma@iis.u-tokyo.ac.jp

Dear Dr. Remias,

   We would appreciate very much a number of valuable comments. Please see enclosed our responses to the all your comments as well as the revised marked-up manuscript entitled as "Temporal changes in snow albedo, including the possible effects of red algal growth, in northwest Greenland, simulated with a physically based snow albedo model" by Yukihiko Onuma et al. [Paper # tc-2019-263] submitted to the journal The Cryosphere. Our responses (**blue text**) to each your comment (**black text**) were described on the following pages. We also uploaded manuscript, which was revised with yellow marker as suggested, on the discussion board.

Best regards,
Yukihiko Onuma and co-authors

General Comments:

   This study links field observations of arctic snow packs at Greenland with phenomena influencing the surface albedo and thus alterations of melting rates with a mathematical model proposed by the authors. The model is intended to explain the role, respectively to forecast the consequences of different surface concentrations mineral dust, black carbon and organic impurities (mainly snow algae) on the seasonal melting rates.

   The questions of this work are of very high ecological relevance, taking global warming into account, and in this special case the fate of the Greenland Ice Sheet. Several important studies recently dealt with albedo changes of bare glacier ice, however partly neglecting the role of decreased albedo of snow on these glaciers, which will lead to earlier exposure of the glacier surfaces in the ablation zone during the melting season.

   The proposed model is feasible and a first good step for performing simulation of scenarios. As for every model, improvements will likely follow to make it more robust, and satellite data generally needs more accurate supplementation of field data acquired in situ. In general, false positive results have to be excluded. In the case of this work, one distinct glacier has been sampled during one season.

We also consider that it is important to understand albedo, including the effect of snow algal blooming, changes of snow on glaciers in Greenland Ice sheet. In order to quantify the snow albedo reduction caused by snow algae using a physically based snow albedo model, we think that the detailed time series observations, including the measurements of snow impurity concentrations, snow physical properties and meteorological conditions, are required first. Of course, observation to quantify snow biological properties should be needed to establish and validate the model. Please see our responses to your comments below.

Major Comments:

1. My review will focus on biological aspects; I am not a specialist for albedo or mathematical simulations. The main issue of this manuscript is that the authors will have to consider the biology of algae causing red snow worldwide in more detail, and as a consequence, the model of snow algae "growth on snow surfaces" (which does practically not take place) has to be modified.

As you pointed out, we should consider the biological aspect (morphology, cell size, pigment and cell movement etc.) more in order to reproduce red snow worldwide using our model of snow algae "growth on snow surfaces". We added the explanation about current status and issues of the model in this manuscript (Lines from 336 to 343 and from 382 to 385).

2. Regarding terminology, "red algae" and "bloomings" should be avoided throughout the

manuscript, and correctly "red snow algae" or better "blooms of red snow" should be used. Consequently, the title could be altered to i.e. "Temporal changes in snow albedo, including the possible effect of algae causing red snow, simulated with a physically based model"

Following your advice, we avoided "red algae" and "bloomings" in the manuscript, so we used "red snow algae" and "blooms of red snow" instead. In addition, our title has been altered to "Temporal changes in snow albedo, including the possible effect of red snow algae, simulated with a physically based snow albedo model" based on your suggestion.

3.  The globally most common algal species causing the well-known red snow phenomenon, also at Greenland, has been described recently as Sanguina nivaloides, and consequently "Chlamydomonas nivalis" should be avoided from now on. The according reference to be incorporated: Procházková et al. 2019 (https://doi.org/10.1093/femsec/fiz064). Likewise, the newest, updated general review for snow algae can be included: Hoham & Remias (https://doi.org/10.1111/jpy.12952; succeeding Hoham & Duval 2001). In these references, the authors can learn that the spherical red cysts which are abundant on snow surfaces, do not cleave at the surface during the melting season. Cell division takes place only early in the season when the population has not yet reached the still white surface, but the bottom snow is already water logged after the end of winter. This has a significant consequence for the proposed "snow algae growth factor": it does not exist! But from where comes the evident increase of seasonal cell concentration of such blooms? The only explanation in my mind is, given that the algae population stays about the same during snowmelt (except that cell diameter will increase to a certain extend), that the volume of snow decreases due to melting (and partly sublimation). Thus, the cell numbers per snow volume increase only passively in the case of red cysts. For creating a model regarding the albedo issues caused by snow algae, the theory can be kept but renamed to i.e. "accumulation model".

We modified the observed algal species in this study to *Sanguina nivaloides* from *Chlamydomonas nivalis* in the manuscript. Also, we referred to the following papers to review the latest biological study in the introduction of the manuscript. We agree that the algal cyst observed in our samples does not increase by cell division. In addition, the increment of cell concentration caused by the decrease of snow volume is would occur as you mentioned. We reported that the red cyst concentration of *Sanguina nivaloides* gradually increased with snow melting on the surface snow in the study site (Onuma et al., 2018). Snowpit observation conducted in the study site showed that the depth of snow was 110 cm when the snow algae first appeared on the snow surface. The snow temperature in the bottom layer (the thickness of 100-110 cm) was -1.7°C, indicating that there was no liquid water at

the layer and thus algal growth at the layer seems to be impossible. In addition, algal cell was not contained in any snow samples collected from the snowpit except the samples at the surface. Nevertheless, the algal cell concentration gradually increased from 30 June to 3 August. Based on these results, we would conclude that the algae grew at the surface snow layers. Of course, we could rarely observe vegetative cells in the snow, thus it is necessary to study further their life cycle on this glacier. Also, it would be possible accumulate and remove of algal cells at the surface. Therefore, we refer the model as "snow algae model", which include both of algal growth and accumulation in this study.   Based on your comments, we have renamed "algal growth model" to "snow algae model" in the whole of the manuscript. Accordingly, we described that the detailed assumption and current issue of snow algae model (Lines from 328 to 355).

Reference:

Hoham, R. W. and Remias, D.: SNOW AND GLACIAL ALGAE: A REVIEW., J. Phycol., doi: 10.1111/jpy.12952, 2020.

Procházková, L., Leya, T., Křížková, H. and Nedbalová, L.: Sanguina nivaloides and Sanguina aurantia gen. et spp. nov. (Chlorophyta): the taxonomy, phylogeny, biogeography and ecology of two newly recognised algae causing red and orange snow., *FEMS Microbiol. Ecol.*, 95:fiz064, doi: 10.1093/femsec/fiz064, 2019.

4.  In this manuscript, the "typical" cell concentration per red snow volume is mentioned several times. But what is typical? First of all, it depends on the location of the ecosystem. Coastal snow fields are usually affected by local nutrient input and may have striking blood red coloration, whereas oligotrophic alpine and polar snow has much lower abundances, causing a more pink snow.

We agree with you. Typical cell concentration for red snow depends on the location of the ecosystem. We think that the study site (Qaanaaq Glacier) fall into the category of oligotrophic polar snow. The information of the location (oligotrophic polar and alpine snow) for definition of "typical red snow" has been added in the manuscript (Line 319). In addition, the range of the typical algal cell concentration in this study has been modified (Line 320). Accordingly, the results in Figures 6 and 7 and the sentences in from Discussion 4.5 to Discussion 4.8 have been modified.

5.  Moreover, the concentration of the organic impurities on the snow surface can also be influenced by meteorological events like precipitation, and the authors should discuss this aspect.

The precipitation event should be discussed to reveal a factor affecting temporal changes in algal abundance. However, we believe that the discussion would be outside the scope of our paper because snow algal abundance did not significantly affect the surface albedo during the period. The OC and algal cell concentrations ranged from $3.2 \times 10^{-2}$ to $1.3 \times 10^{-1}$ mg $L^{-1}$ and 0 to $1.3 \times 10^{-4}$ cells $L^{-1}$ in the surface snow from day 168 to day 209, respectively, indicating that the concentrations are pretty lower than that of typical red snow in oligotrophic polar snow. The discussion about the temporal changes in the lower algal cell concentration may lead to confuse the issue (i.e. contribution of red snow blooming to snow albedo). Therefore, we did not add the discussion to the manuscript in this time although precipitation event is an important factor to discuss temporal changes in snow algal abundance.

6. Melting surface snow has temperatures about 0.5 to 1 C in all studies I know. Therefore, why were calculations performed simulating an elevated snow temperature of +1.5C? This should be physically impossible, and thus I suggest removing this hypothetical data from the manuscript, and likewise change fig. 7.

In our simulation, snow temperature becomes 0°C when the temperature exceeds 0°C by warming test. Our explanation in the manuscript was insufficient about that, so the explanation has been described (Lines from 445 to 446).

Detail comments:
1. Line 16: use Sanguina nivaloides instead of Chlamydomonas nivalis throughoutge'
The species name has been corrected in the whole of the manuscript.

2. Line 27: Why should this albedo model only be valid for snow covers in Greenland? Could it be applied elsewhere?
The albedo model was established on the basis of the observation in Qaanaaq Glacier. Unfortunately, we have only the validation data in the study site (snow impurity concentrations and snow physical properties in surface and subsurface snow layers). However, we plan to apply the model to the other site (for example, Alaskan glacier and Svalbard). The current issue and future plan have been described in the manuscript (Lines from 381 to 385).

3. Line 41: visible is from 400 to 700 nm
The word has been corrected (Line 42).

4. Line 47: "of" instead "in" the Greenland Ice Sheet

The word has been corrected (Line 48).

5.  Line 50: insert "algae" to the list in brackets

The word has been inserted (Line 52).

6.  Line 52: insert "in" after "might be present"

Following another reviewer's comment, "on" has been inserted after "present" (Line 52).

7.  Line 57: Start sentence with "The …"

The word has been inserted (Line 58).

8.  Line 67: reference Yallop et al. 2012 is about glacier algae, not about snow algae

The reference has been removed (Line 69).

9.  Line 68: delete "and ice" for snow algae.

The word has been deleted (Line 70).

10. Line 70: give reference Hoham & Remias instead of Hoham & Duval

The reference has been replaced with Hoham & Remias (Line 71).

11. Lnie 71: use Sanguina nivaloides instead of Chlamydomonas nivalis and include reference
    Procházková et al 2019

The sentence has been modified as suggested. In addition, the reference has been included in the sentence (Lines from 72 to 75).

12. Line 74: "algal blooms have" instead of "algal blooming has"

The words have been corrected (Line 76).

13. Line 78: "cellular pigment composition", delete "in cell"

The words have been corrected (Line 80).

14. Line 84 & 93: reformulate as explained earlier: the algal abundance on surface is not due to growth
    but due to concentration. E.g. "accumulation model"

As we mentioned at Major comment 3, our model considers both effects of algal growth and accumulation to reproduce algal cell abundance. We have modified the sentence (Lines from 86 to 87).

15. Line 191: The mean cell radius was 11.4 µm – based on algae from the study site? Please state. Elsewhere in the manuscript, a different cell size number based on an older study of Onuma et al. was given.

Yes, the size was measured using the cells observed in the study site. Onuma et al. (2018) showed that the mean cell size at the study sites (Sites A and B in the previous study), but we showed that the mean cell size at Site B only in this study. The explanation has been added to the manuscript (Line198).

16. Line 237: the decrease of OC on days 197 and 209 should be discussed, and please consider meteorological events like precipitation as a causer?

As we answered at Major comment 5, we did not add the discussion to the manuscript in this time because we believe that the discussion would be outside the scope of our paper.

17. Lines 240 – 245: The correlation between OC concentration and snow algae abundance appears to be obvious. Still, despite high significance, other organisms can also contribute to OC like bacteria, and in this habitat most likely, yeast-like fungi. This should be at least briefly discussed (e.g. line 285: the formula is not a proof that snow algae are the main constitute of OC in snow – though this is apparent for this study). Since fungal blooms depend on the snow algae, there could be a stable correlation as well.

Although we could not quantify the abundance of fungi, we have quantified the abundance of *Chroococcaceae cyanobacterium* during the observational period. The abundance ranged from 0 to $9.7 \times 10^5$ cells m$^{-2}$ from day 168 to day 215, so it possibly contributes to OC concentration. However, the cell size was very smaller ($2.3 \pm 0.6$ µm in radius) compared with that of red cyst of *Sanguina nivaloides*. Therefore, we assumed that the effect of the cyanobacteria can be neglected to obtain the relationship between the red cyst abundance of *Sanguina nivaloides* and the concentration of OC. As you pointed out, we should discuss the contribution of bacteria and fungi to OC concentration. The discussion has been described (Lines from 299 to 304).

18. Line 285: "constituent" instead of "constitution"

The word has been corrected (line 295).

19. Line 289: please explain/hypothesise how algae-free snow contains significant amount of OC at a remote place of Northwestern Greenland. Long-distance deposition?

Following the previous your comment (Detail comments 17), we reconsidered the reason for the intercept of 0.0826 of equation (3) in the manuscript. Probably, the contribution of cyanobacteria abundances is included in the intercept value. The discussion about the reason of the intercept value has been described (lines from 299 to 304).

20. Line 296: unneeded repetition of the considered pigments

The words have been deleted (Line 307).

21. Line 297: "absorption", not "abruption"

The word has been corrected (Line 307).

22. Line 299: Sentence "The dominant species was …" is a repetition of results.

The sentence has been removed (Line 310). Also, sentence "These results suggest" has been corrected (Line 310).

23. Line 309: the typical cell concentration of the study site was 4.9 x 10 e4 cells per litre. To my experience, the lower threshold to see snow discolorations by one's eye is about 5 x 10 e6. Thus my question, did you see red snow visually at the study site? This should be mentioned in the results.

We could not see red snow visually at the study site in 2014 season. However, red snow appeared visibly on surface snow at the lower site of Qaanaaq Glacier when the cell concentration of the red cyst of *Sanguina nivaloides* was $3.2 \times 10^6$ cells $L^{-1}$ (Onuma et al., 2018). To our knowledge, the cell concentration was slightly lower than the concentration in the other sites reported by previous studies. So, we assumed that the minimum value of typical red snow is $3.2 \times 10^6$ cells $L^{-1}$ in this study. The explanation has been added (Lines from 319 to 322).

24. Line 310: delete non-existing reference "Sutton et al. 1972". There exists only a PhD of Mrs. Sutton from that year, a fine work though which was never published as a paper. There are many good papers for the concentration of Sanguina nivaloides in red snow.

The reference has been deleted, so some reference (Takeuchi and Koshima, 2004; Takeuchi et al., 2006; Stibal et al., 2007; Tanaka et al., 2016; Procházková et al., 2018) have been added in the manuscript instead (Lines from 320 to 321). The range of the typical red snow concentration has been modified (Line 320). Also, the snow albedos were recalculated using the newer range of the algal cell concentration with PBSAM. Accordingly, the results in Figures 6 and 7 and the sentences in from Discussion 4.5 to Discussion 4.8 have been modified.

Reference:
Takeuchi, N. and Kohshima, S.: snow algal community on a Patagonian glacier, Tyndall glacier in the Southern Patagonia Icefield, *Arct. Antarct. Alp. Res.*, 36, 91–8, 2004.

Takeuchi, N., Dial, R., Kohshima, S., Segawa, T. and Uetake, J.: Spatial distribution and abundance of red snow algae on the Harding Icefield, Alaska derived from a satellite image. *Geophys. Res. Lett.*, 33, L21502, doi: 10.1029/2006GL027819, 2006.

Stibal, M., Elster, J., Ŝabacká, M. and Kaŝtovská, K.: Seasonal and diel changes in photosynthetic activity of the snow alga Chlamydomonas nivalis (Chlorophyceae) from Svalbard determined by pulse amplitude modulation fluorometry. *FEMS. Microbiol. Ecol.*, 59, 265–273, doi:10.1111/j.1574-6941.2006.00264.x, 2007.

Tanaka, S., Takeuchi, N., Miyairi, M., Fujisawa, Y., Kadota, T., Shirakawa, T., Kusaka, R., Takahashi, S., Enomoto, H., Ohata, T., Yabuki, H., Konya, K., Fedorov, A., Konstantinov, P.: Snow algal communities on glaciers in the Suntar-Khayata Mountain Range in eastern Siberia, Russia. *Polar Sci.*, 10, 3, 227-238, doi: 10.1016/j.polar.2016.03.004, 2016.

Procházková, L., Remias, D., Holzinger, A., Řezanka, T. and Nedbalová, L.: Ecophysiological and morphological comparison of two populations of Chlainomonas sp. (Chlorophyta) causing red snow on ice-covered lakes in the High Tatras and Austrian Alps., *Eur. J. Phycol.*, 53:230–43, doi:10.1080/09670262.2018.1426789, 2018.

25. Chapter 4.3 (lines 316 – 337): I have serious problems with this part. It should either be rewritten or omitted. There are wrong biological assumptions about growth of red snow algal cysts (as stated in the beginning), which simply does not take place on the snow surface. The bloom that you can see is a (physiologically active) resting stage.

In this study, cell concentration of red cyst of *S. nivaloides* was calculated with "snow algae model". As we mentioned at Major comment 3, we have described the detailed assumption and current issue of the snow algae model (Lines from 336 to 358).

26. Chapter 4.4: A "typical" red snow bloom should be defined more clearly. Only two references for average cell numbers are given. Furthermore, Painter et al 2001 is inappropriate in this context, because they clearly state that they did not any field measurements. Moreover, the number taken from Lutz et al 2014 from location MIT-17 is not the average value for red cysts in that study, but a rather low value for a bloom. For being representative, I strongly recommend to include certain further studies of "Chlamydomonas nivalis" and then recalculate the albedo reduction of a "typical" red bloom.

As we mentioned at Major comment 4, the definition of "typical red snow" has been added in the manuscript (Line 319). And, we have modified the range of "typical red snow" in the manuscript.

Accordingly, the snow albedos were recalculated using the newer range of the algal cell concentration with PBSAM. Based on your comment, we have added model validations with two field observation data (Takeuchi et al., 2006 and Takeuchi, 2013). In this study, we used only field observation data collected by Painter et al. (2001, Fig. 1). To our understanding, the data of the figure 1 seemed to be obtained from field measurement and sampling although the other data were not obtained from those. In Lutz et al. (2014), the snow albedo in the visible band and algal cell concentration were observed simultaneously at Mit-17, so we consider that the model validation can be conducted using the observational data. To explain clearly, we have summarized the information about these model validations as a table (Lines from 357 to 359 and Table 2). Based on these validations, we have discussed about possible biological factors affecting snow albedo (cell size, algal community and pigment composition) (Lines from 362 to 385).

27. Line 353: Start sentence "The phenolic pigments of glacier algae have a broader bandwidth of spectral absorption than the carotenoids of S. nivaloides". Instead of citing Dial et al. 2018, which is rather a deductive/theoretical work, a reference showing real spectral data is more appropriate (e.g. Fig. 13.6 in Remias 2012, Springer Vienna)

The sentence has been modified as suggested (lines from 372 to 374). And, the following references have been added in the manuscript.

Reference:

Remias, D.: Cell structure and physiology of alpine snow and ice algae, in: Plants in alpine regions, Cell physiology of adaption and survival strategies, edited by: Lütz, C., *Springer Wien*, 202, 175–186, doi: 10.1007/978-3-7091-0136-0_13, 2012.

Williamson, C. J., Cook, J. M., Tedstone, A., Yallop, M., McCutcheon, J., Ponieckae, E., Campbell, D., Irvine-Fynn, T., McQuoid, J., Tranter, M., Perkinse, R. and Anesio, A.: Algal photophysiology drives darkening and melt of the Greenland Ice Sheet, *PNAS*, 117, 11, 5694-5705, doi:10.1073/pnas.1918412117, 2020.

28. Line 363: Again, Sutton 1972 is not a paper and Painter et al 2001 give no numbers to cite for.

These reference have been deleted, so some reference (Takeuchi and Koshima, 2004; Takeuchi et al., 2006; Stibal et al., 2007; Tanaka et al., 2016; Procházková et al., 2018) have been added in the manuscript instead (Lines from 391 to 392).

29. Line 388: replace "but" with "and"

The word has been corrected (Line 417).

30. Chapter 4.8: Needs to be updated. Please think over the "snow algae growth model" and the sense of using hypothetical high snow temperatures for any calculations.

As we mentioned at Major comments 3, these explanations have been added to the discussion part in the manuscript (Lines from 328 to 355).

31. Growth of snow algae takes place in deep layers of snow with very constant conditions around 0.5°C. The air temperature plays generally no big role since the snow pack starts to be water logged from the bottom, and this occurs earlier than air temperatures raise above zero.

We also think that melting water is an important factor affecting algal growth. Onuma et al. (2018) reported that cells of *S. nivaloides* were not observed from bottom to surface in snowpack at this study site on mid-June in 2014. This is probably due to existence of super imposed ice on the bottom layer. In the case, the red algal cells appear to have originated from windblown algal spores in the atmosphere, but they are not likely from the remaining snow of the previous melt season. Moreover, snow surface temperature was almost 0°C at the study site from late June to early August in 2014. Logistic model may reproduce the temporal change in algal abundance under such conditions. Because the explanation has been described in Onuma et al. (2018), the brief explanation has been added to our manuscript as suggested (Lines from 328 to 355).

32. Line 586: Reference Painter et al 2001 incomplete, authors missing. "Thomas" instead of Thimas".

The author name has been corrected (Line 622).

33. Legend to fig 1: "sampling site", not "sites"

The word has been corrected (Fig. 1).

34. Legend to fig. 5: say "Temporal changes in algal cell concentration and …"

The words have been replaced as suggested (Fig 5).

35. Figure 7: This is a bit vague considering the theory of cell growth and snow temperatures of +1.5°C

In our snow algae model, snow algal cell concentration gradually increased with snow melting, which means that snow temperature on surface exceed 0°C. And, snow temperature becomes 0°C when the temperature exceeds 0°C by the warming test. Therefore, the sensitivity test of temporal changes in the algal cell abundance can be conducted with the snow algae model using different snow temperature conditions. The explanation has been described (Lines from 445 to 446).

---

## Author Comment (AC2) · 13 Apr 2020

**Reply to Dr. Marian Yallop (Reviewer#2)**

April 3, 2020
Dr. Yukihiko Onuma
Institute of Industrial Science, University of Tokyo
E-mail: onuma@iis.u-tokyo.ac.jp

Dear Dr. Yallop,

We would appreciate very much a number of valuable comments. Please see enclosed our responses to the all your comments as well as the revised marked-up manuscript entitled as "Temporal changes in snow albedo, including the possible effects of red algal growth, in northwest Greenland, simulated with a physically based snow albedo model" by Yukihiko Onuma et al. [Paper # tc-2019-263] submitted to the journal The Cryosphere. Our responses (**blue text**) to each your comment (**black text**) were described on the following pages. We also uploaded manuscript, which was revised with yellow marker as suggested, on the discussion board.

Best regards,
Yukihiko Onuma and co-authors

**General comments:**

   The role played by snow algae in contributing to albedo is currently of relevance and likely to be of interest to the larger cohort of scientists working in this field. This paper describes seasonal changes in the 'cyst stage' of a species they have identified as mostly, 'Chlamydonomas nivalis' type cells, documenting changes in numbers with time and modelling the contribution these cells make to a physically based snow albedo model. This species has now been renamed as Sanguina nivaloides (Procházková et al. 2019. The measurements have been made on a snowpack in northwest Greenland during the ablation period of 2014. Samples were also collected to quantify mineral particles and organic and black carbon content. However, an assumption is made that the cells are actively growing on the ice, i.e. that cell division leads to an increase in numbers through time. However, the cysts do not divide on the surface of the snow, but will, once buried, potentially provide the inoculum for the following year should suitable conditions arise. In the spring or early summer, once melt water forms, the flagellated algal cells will swim up to the snow surface to form a bloom. On the surface, they lose their flagella and formation of the red pigments occur. Members of the so-called water-melon snow are likely more speciose than was once thought. The development of blooms on the surface can show considerable spatial and temporal heterogeneity, at any one point in time, making it difficult to reliably quantify their distribution. This patchiness may be due to species-specific pigment differences as well as change in their relative abundance, and these factors needs to be carefully considered for inclusion in any model where the aim is to quantify their contribution to albedo. However, I do not have expertise in the modelling sections and my comments refer to the biological sections of the manuscript.

   As you pointed out, we have to establish microbial and albedo models based on an appropriate biological assumption. Previous study (Onuma et al., 2018), we observed temporal changes in algal cell concentration of *Sanguina nivaloides* on surface snow in Qaanaaq Glacier and reproduced the temporal change with a Logistic model, but the cells observed were mostly cyst. As cyst does not increase by cell division, the expression of "algal growth" may inappropriate for the model. However, snowpit observation conducted in the study site showed that the depth of snow was 110 cm when the snow algae first appeared on the snow surface. The snow temperature in the bottom layer (the thickness of 100-110 cm) was -1.7°C, indicating that there was no liquid water at the layer and thus algal growth at the layer seems to be impossible. In addition, algal cell was not contained in any snow samples collected from the snowpit except the samples at the surface. Nevertheless, the algal cell concentration gradually increased from 30 June to 3 August. Based on these results, we would conclude that the algae grew at the surface snow layers. Of course, we could rarely observe vegetative cells in the snow, thus it is necessary to study further their life cycle on this glacier. Also, it would be possible accumulate and remove of algal cells at the surface. Therefore, we refer the model as "snow algae model", which include both of algal growth and accumulation in this study. Based on your comments, we have

renamed "algal growth model" to "snow algae model" in the whole of the manuscript. Accordingly, we described that the detailed assumption and current issue of snow algae model (Lines from 336 to 343). Also, we have modified the algal species name (*Chlamydomonas nivalis*) to *Sanguina nivaloides* in the whole of the manuscript (Procházková et al., 2019 has been added as a reference).

Regarding to the life cycle of *Sanguina nivaloides* in snowpack, the cells were not observed from bottom to surface in snowpack at this study site on mid-June in 2014 reported by Onuma et al. (2018). This is probably due to existence of super imposed ice on the bottom layer. In the case, the red algal cells appear to have originated from windblown algal spores in the atmosphere, but they are not likely from the remaining snow of the previous melt season. Because the discussion has been described in Onuma et al. (2018), the brief explanation has been added to our manuscript only (Lines from 336 to 343).

Regarding to spatial heterogeneity of the blooms, we used spatial mean of snow impurity concentrations and snow physical properties at the study site as input variables in order to simulate temporal changes snow albedo including the effect of red snow. The model parameters required by "snow algae model" in this study were obtained from spatial mean cell concentration observed at the site. Therefore, we focused on temporal change in "site mean" snow albedo in this study. The explanation has been added (Line 127). Unfortunately, we did not conduct the detailed spatial observation (the spatial distribution of algal cell concentration and snow albedo). Because the spatial heterogeneity of red snow is important to estimate accurately snow melting with an albedo model, we have added the point as a future task in the manuscript (Lines from 383 to 385).

Reference:

Procházková, L., Leya, T., Křížková, H. and Nedbalová, L.: Sanguina nivaloides and Sanguina aurantia gen. et spp. nov. (Chlorophyta): the taxonomy, phylogeny, biogeography and ecology of two newly recognised algae causing red and orange snow., *FEMS Microbiol. Ecol.*, 95:fiz064, doi: 10.1093/femsec/fiz064, 2019.

**Major Comments:**

It is recommended that consideration be given to rewording the title for two reasons: i) the named algae are green algae, members of the Chlorophyta or green algae, and readers may be confused into thinking that this paper is about red algae (Rhodophyceae); ii) these particular algae are not considered to grow (as in cell division) once on the surface; they rapidly form cysts, though the cysts themselves may potentially show size changes through time, and they can still be photosynthesising but at very low rates i.e. they can still be metabolically active if the conditions are right. But, it is these cysts that will eventually act as the inoculum for the next year hence their persistence on the surface is fundamental for species survival from year to year. They also form an important food source for a

number of grazers. Through the text, including the abstract, it is important that the term red algae is revised accordingly. It needs to be emphasised that the cells, on reaching the surface would turn into cysts and that an increase in the concentration of cell numbers on the surface is as a result of cell accumulation as the snow melts and algae concentration as they get behind.

For this reason, all reference to algal growth must be removed. That said, it is possibly that some other microbes may be found growing on the snow but the particular species mentioned here would not. The researchers may find these additional papers of use. Fogg (1967) Phil. Trans. Soc , 252, 279-287. Fjerdingstad et al., 1974, Arch. Hydrobiol. 73, 70-83 as well as the newer review by Hoham & Remias, Journal of Phycology, 2020. Line 27, clarify what is meant by proper estimates?

Many references are made in relation to the growth of the snow algae on the surface, as the snow is melting. It would be very useful to add in more detail about the life cycle of the collective group of snow algae, detailing the light triggers that promote the of the biflagellate stage to the ice surface and their modification to form the resulting cyst stage, with loss of the flagellae. The transition period from one form to another, and the time period over which this may happen, is critical. Fogg et al. (1974) report that the increase in cell numbers in snow is sometimes as a result of cell concentration due to ablation (sublimation) which leaves the algae behind. Previous researchers report on active photosynthesis in these surface cells, though possibly activity would cease once the surface temperatures become too high. The review of Hoham and Remias, (2020) would also be useful here.

We have incorporated your suggestions. First, the title were altered to "Temporal changes in snow albedo, including the possible effect of red snow algae, simulated with a physically based snow albedo model" according to your suggestion. We agree with your biological suggestion. Unfortunately, we did not conduct the detailed biological observation to quantify temporal changes in life stage, photosynthesis and metabolically active of *Sanguina nivaloides*. So, we have described the current status and issue of our snow algae model in the manuscript (Lines from 336 to 343). Because many reference paper in the manuscript have reported the temporal changes in algal abundance, which was resulted from "algal growth and/or accumulation", those have remained in the manuscript. Also, the following references have been added to the manuscript as suggested.

References:

Remias, D, Cell structure and physiology of alpine snow and ice algae, in: Plants in alpine regions, Cell physiology of adaption and survival strategies, edited by: Lütz, C., *Springer Wien*, 202, 175–186, doi: 10.1007/978-3-7091-0136-0_13, 2012.

Procházková, L., Remias, D., Holzinger, A., Řezanka, T. and Nedbalová, L., Ecophysiological and morphological comparison of two populations of Chlainomonas sp. (Chlorophyta) causing red snow on ice-covered lakes in the High Tatras and Austrian Alps., *Eur. J. Phycol.*, 53:230–43,

doi:10.1080/09670262.2018.1426789, 2018.

Hoham, R. W. and Remias, D.: SNOW AND GLACIAL ALGAE: A REVIEW., J. Phycol., doi: 10.1111/jpy.12952, 2020.

**Suggest revise wording:**

1. Line 33, add Yallop et al., 2012 to the references reporting changes in surface albedo with ice melt;

The reference has been added (Line 34).

2. Line 41 – check all references made to ranges for visible light as they vary through the text;

The wavelength in the visible band has been corrected (400-700 nm, Line 42).

3. Line 45 – What is meant by ppbw (provide full definition);

ppbw is an abbreviation of parts per billion weight, a subunit of ppb that is used for part of weights like micrograms per kilogram (μg/kg). To explain simply, we modified the unit to μg/kg from ppbw (Line 46).

4. Line 46 Suggested reordering of this sentence: : : :.. was reported to be lower, by 0.7%, than that by BC: : :;

To explain clearly, the value (0.7%) has been changed to 'approximately 1%' (Line 47).

5. Line 49 – change absorb to absorbs;

The word has been corrected (Line 50).

6. Line 51 – after present add an on;

The word has been added (Line 52)

7. Line 57 Revise to read A physically based snow : : :: : :.;

Following the another reviewer's comment, the define article has been added to the sentence beginning (Line 58).

8. Line 83 – and elsewhere – revise references to 'growth' of cells on the surface and references to growth model where the 'growth' is likely resulting from an accumulation of cells in a defined area rather than active division of cells.

As we mentioned at Major comment, we have added the more detailed assumption of snow algal

growth (Lines from 336 to 343).

9. Line 65, Yallop et al 2012 discussed potential ice algal albedo impacts not snow;
The reference has been removed (Line 69).

10. Line 96, Field not filed;
The word has been corrected (Line 100).

11. Line 103 – through more recent molecular work it is likely that any surface blooms may contain a number of different species that have very similar morphologies and it may be better to use Chlamydomonas spp., to infer that. There is also an indication that there were some cells that were not spherical red cells. Can any more information be provided regarding the identity of these cells?;
Unfortunately, the detailed molecular work has not been conducted in this study. Our study focus on the contribution of red snow to snow albedo, so we consider that the detailed discussion would be outside the scope of our paper. However, the point is important to reveal the contribution of snow algae to snow albedo. We have added the future plan in the manuscript (Lines from 381 to 383).

12. Line 116. It would be useful to add more information about the density samples (make, model);
In this study, we used the snow density as input variable for PBSAM, so the snow properties were observed. The objective of the measurement has been described (Lines from 122 to 123). And, the information of the density sampler has been added (Line 120).

13. Line 120 . It is recommended that more information is provided in the text to provide details about the spatial sampling protocol.
The snow sample corrections have been conducted in about 15x15 m spatial scale. The information has been added (Lines from 126 to 127).

14. Line 135 – change bag to bags;
The word has been corrected (Line 144).

15. Line 145 – after USA), change the 'in' to an 'on';
The word has been replaced (Line 153).

16. Line 145 – were the samples preserved or rather maintained. Using the term may imply some preservative was added.

The word has been changed to 'maintained' as suggested (Line 153).

17. Line 192 cell sizes were measured. Do these sizes account for any shrinkage as a result of the preservative used?;

In this study, cells obtained from snow sample stored in a freezer were used to measure the cell size. Unfortunately, we did not measure such shrinkage in this study. However, the effect of the cell shrinkage on light absorption in snowpack should be considered, so the information of the counted cell has been described (Line 199).

18. Line 194 and 195– remove the 's' from compositions;

The 's' has been removed (Line 203).

19. Line 195 – revise sentence as meaning is not clear;

As you know, we assumed four light absorption pigments (chlorophyll-a, chlorophyll-b, primary carotenoids, and secondary carotenoids) to simulate light absorption caused by red snow in this study. Also, we assumed the concentration of each pigment in the algal cells were based on Cook et al. (2017, JGR). However, the information was missing. The explanation has been described (Lines from 204 to 205).

20. Line 245 – Why is there a – sign in the equation for cell numbers?;

To prepare an input data for PBSAM, the concentration of algal cells was converted into that of OC using the equation. To explain clearly, multiplication symbol has been inserted between '$5.3 \times 10^{-6}$' and '$C_{algae}$' (line 255). In addition, the unit for '$C_{oc}$' has been added (Line 256).

21. Line 261: Relevant to this comments is the potential for aggregation of material on the snow. The cells can be sticky and aggregate to form larger clumps, together with mineral particles and other associated matter, including bacteria. Aggregation may affect their motility.

Temporal changes in MD and algal cell concentrations in surface and subsurface snow gradually increased from late June to early August in 2014 at the study site. These results suggest that such aggregation did not significantly affect temporal change in algal cell concentration in the case of this study. Because we have already discussed about the effect of MD concentration on algal growth on surface snow in previous study (Onuma et al., 2018), we believe that the discussion would be outside the scope of our paper.

22. Line 285 – change the word 'constitution' for 'constituent';

The word has been corrected (Line 295).

23. Line 268 – change 'amount' to 'amounts';

The word has been corrected (Line 298).

24. Line 309, here and elsewhere in text, if the cell numbers being reported are averages, can the SD or SE of cell number be added ;

The cell numbers were obtained from three snow samples, which were collected from three surfaces selected randomly at the study site. The standard deviation has been added to the sentence as suggested (Line 318). In addition, we have added "(mean ± SD)" in the result section 3.2 (Line 236).

25. Line 351 more literature and information could be added to support this statement that there may be different pigments in ice surfaces. The authors are referred to the new paper by Williamson et al. 2020 (PNAS , www.pnas.org/cgi/doi/10.1073/pnas.1918412117), for further views on the role of pigments. Importantly, whilst it is possible that snow algae may be found in ice environments, some of the major players on the ice sheet e.g. Ancylonema may not grow on the snow. Further, the latter species is actively growing and not in a resting stage hence it might be expected that their pigments would be very different.

We agree with your suggestion. The effect of algal pigment and species on surface albedo should have been discussed more detailed. We have added the following paper as references in the manuscript. And, the discussion about the effect of algal pigment and species on the snow albedo has been added (Lines from 372 to 374).

References:

Williamson, C. J., Cook, J. M., Tedstone, A., Yallop, M., McCutcheon, J., Ponieckae, E., Campbell, D., Irvine-Fynn, T., McQuoid, J., Tranter, M., Perkinse, R. and Anesio, A.: Algal photophysiology drives darkening and melt of the Greenland Ice Sheet, PNAS, 117, 11, 5694-5705, doi:10.1073/pnas.1918412117, 2020

Remias, D.: Cell structure and physiology of alpine snow and ice algae, in: Plants in alpine regions, Cell physiology of adaption and survival strategies, edited by: Lütz, C., Springer Wien, 202, 175–186, doi: 10.1007/978-3-7091-0136-0_13, 2012.

26. Line 410 onwards – references of growth of snow algae need to be removed here, though some snow algae may have vegetative stages in the snow.

As we mentioned at Major comment, we have described the detailed explanation of "algal growth" in the manuscript. Because Onuma et al. (2018) has reported the temporal changes in algal abundance of

red cyst (*Sanguina nivaloides*), which was resulted from "algal growth and/or accumulation", it has remained in the manuscript. However, another one (Onuma et al. 2016) has been removed due to research on *Chloromonas nivalis* (Line 440).

---

## Author Comment (AC3) · 14 Apr 2020

**Reply to Dr. Daniel Remias (Reviewer#1)**

April 14, 2020
Dr. Yukihiko Onuma
Institute of Industrial Science, University of Tokyo
E-mail: onuma@iis.u-tokyo.ac.jp

Dear Dr. Remias,

We would appreciate very much a number of valuable comments. Please see enclosed our responses to the all your comments as well as the revised marked-up manuscript entitled as "Temporal changes in snow albedo, including the possible effects of red algal growth, in northwest Greenland, simulated with a physically based snow albedo model" by Yukihiko Onuma et al. [Paper # tc-2019-263] submitted to the journal The Cryosphere. Our responses (**blue text**) to each your comment (**black text**) were described on the following pages. We also described major revised sentences based on your suggestion (revised part: yellow marker), after our response.

Best regards,
Yukihiko Onuma and co-authors

General Comments:

This study links field observations of arctic snow packs at Greenland with phenomena influencing the surface albedo and thus alterations of melting rates with a mathematical model proposed by the authors. The model is intended to explain the role, respectively to forecast the consequences of different surface concentrations mineral dust, black carbon and organic impurities (mainly snow algae) on the seasonal melting rates.

The questions of this work are of very high ecological relevance, taking global warming into account, and in this special case the fate of the Greenland Ice Sheet. Several important studies recently dealt with albedo changes of bare glacier ice, however partly neglecting the role of decreased albedo of snow on these glaciers, which will lead to earlier exposure of the glacier surfaces in the ablation zone during the melting season.

The proposed model is feasible and a first good step for performing simulation of scenarios. As for every model, improvements will likely follow to make it more robust, and satellite data generally needs more accurate supplementation of field data acquired in situ. In general, false positive results have to be excluded. In the case of this work, one distinct glacier has been sampled during one season.

We also consider that it is important to understand albedo, including the effect of snow algal blooming, changes of snow on glaciers in Greenland Ice sheet. In order to quantify the snow albedo reduction caused by snow algae using a physically based snow albedo model, we think that the detailed time series observations, including the measurements of snow impurity concentrations, snow physical properties and meteorological conditions, are required first. Of course, observation to quantify snow biological properties should be needed to establish and validate the model. Please see our responses to your comments below.

Major Comments:

1. My review will focus on biological aspects; I am not a specialist for albedo or mathematical simulations. The main issue of this manuscript is that the authors will have to consider the biology of algae causing red snow worldwide in more detail, and as a consequence, the model of snow algae "growth on snow surfaces" (which does practically not take place) has to be modified.

As you pointed out, we should consider the biological aspect (morphology, cell size, pigment and cell movement etc.) more in order to reproduce red snow worldwide using our model of snow algae "growth on snow surfaces". We added the explanation about current status and issues of the model in this manuscript (Lines from 336 to 343 and from 382 to 385).

Lines from 328 to 343 in the revised manuscript:

Although our field observations ended on day 215, snow algal abundance could further increase until the end of the melting season. In order to infer temporal changes in snow albedo for the whole melting season, we calculated snow albedo using the PBSAM and a snow algae model proposed by Onuma et al. (2018). Temporal changes in abundance of *S. nivaloides* on surface snow of Qaanaaq Glacier can simply be expressed by a differential logistic growth equation. Microbial growth was therefore calculated as follows (Onuma et al., 2018):

$$X = \frac{K}{1 + \frac{K - X_0}{X_0} e^{\mu(t_0 - t)}} \; , \; t = d - \; d_f \tag{4}$$

where $X$ and $X_0$ are population densities of microbes at $t$ and $t_0$, respectively, and $\mu$ is the growth rate of microbes in $t^1$. $K$ is the carrying capacity of algae in the snow surface and $t_0$ is the day of the first appearance of algae on the snow surface. $t$ represents the number of the days during which the snow surface temperature was above 0°C, because snow algal growth mainly occurs on the melting snow surface. Although this model assumes algal growth on the snow surface, the algal cells observed in the surface snow were mostly cyst stage, which does not divide and thus not activity increase their population. The algae may divide at the subsurface or deeper layers in the snowpack. Therefore, the increase of algal cells at snow surface may due to not only their growth but also to accumulation at the surface as snow melt. However, their actual life cycle is still uncertain on this glacier. In this study, we use this model, which may include growth and/or accumulation of the algal cells but can reasonably reconstruct the observation of their seasonal change on the snow surface of the study site (Onuma et al., 2018).

Lines from 381 to 385 in the revised manuscript:

Unfortunately, we have only the validation data in the study site (MD, BC and OC concentrations and snow physical properties in surface and subsurface snow layers). The detailed time series observation, including analysis of cell size, pigment composition, algal community, should be conducted in other sites to evaluate our albedo model. Moreover, the detailed spatial measurements of algal cell abundance and snow albedo would also be needed because patchy distribution of red snow often appear on oligotrophic polar and alpine snow.

2. Regarding terminology, "red algae" and "bloomings" should be avoided throughout the manuscript, and correctly "red snow algae" or better "blooms of red snow" should be used. Consequently, the title could be altered to i.e. "Temporal changes in snow albedo, including the possible effect of algae causing red snow, simulated with a physically based model"

Following your advice, we avoided "red algae" and "bloomings" in the manuscript, so we used "red snow algae" and "blooms of red snow" instead. In addition, our title has been altered to "Temporal

changes in snow albedo, including the possible effect of red snow algae, simulated with a physically based snow albedo model" based on your suggestion.

3.  The globally most common algal species causing the well-known red snow phenomenon, also at Greenland, has been described recently as Sanguina nivaloides, and consequently "Chlamydomonas nivalis" should be avoided from now on. The according reference to be incorporated: Procházková et al. 2019 (https://doi.org/10.1093/femsec/fiz064). Likewise, the newest, updated general review for snow algae can be included: Hoham & Remias (https://doi.org/10.1111/jpy.12952; succeeding Hoham & Duval 2001). In these references, the authors can learn that the spherical red cysts which are abundant on snow surfaces, do not cleave at the surface during the melting season. Cell division takes place only early in the season when the population has not yet reached the still white surface, but the bottom snow is already water logged after the end of winter. This has a significant consequence for the proposed "snow algae growth factor": it does not exist! But from where comes the evident increase of seasonal cell concentration of such blooms? The only explanation in my mind is, given that the algae population stays about the same during snowmelt (except that cell diameter will increase to a certain extend), that the volume of snow decreases due to melting (and partly sublimation). Thus, the cell numbers per snow volume increase only passively in the case of red cysts. For creating a model regarding the albedo issues caused by snow algae, the theory can be kept but renamed to i.e. "accumulation model".

We modified the observed algal species in this study to *Sanguina nivaloides* from *Chlamydomonas nivalis* in the manuscript. Also, we referred to the following papers to review the latest biological study in the introduction of the manuscript. We agree that the algal cyst observed in our samples does not increase by cell division. In addition, the increment of cell concentration caused by the decrease of snow volume is would occur as you mentioned. We reported that the red cyst concentration of *Sanguina nivaloides* gradually increased with snow melting on the surface snow in the study site (Onuma et al., 2018). Snowpit observation conducted in the study site showed that the depth of snow was 110 cm when the snow algae first appeared on the snow surface. The snow temperature in the bottom layer (the thickness of 100-110 cm) was -1.7°C, indicating that there was no liquid water at the layer and thus algal growth at the layer seems to be impossible. In addition, algal cell was not contained in any snow samples collected from the snowpit except the samples at the surface. Nevertheless, the algal cell concentration gradually increased from 30 June to 3 August. Based on these results, we would conclude that the algae grew at the surface snow layers. Of course, we could rarely observe vegetative cells in the snow, thus it is necessary to study further their life cycle on this glacier. Also, it would be possible accumulate and remove of algal cells at the surface. Therefore, we

refer the model as "snow algae model", which include both of algal growth and accumulation in this study. Based on your comments, we have renamed "algal growth model" to "snow algae model" in the whole of the manuscript. Accordingly, we described that the detailed assumption and current issue of snow algae model (Lines from 328 to 343).

Reference:

Hoham, R. W. and Remias, D.: SNOW AND GLACIAL ALGAE: A REVIEW., J. Phycol., doi: 10.1111/jpy.12952, 2020.

Procházková, L., Leya, T., Křížková, H. and Nedbalová, L.: Sanguina nivaloides and Sanguina aurantia gen. et spp. nov. (Chlorophyta): the taxonomy, phylogeny, biogeography and ecology of two newly recognised algae causing red and orange snow., *FEMS Microbiol. Ecol.*, 95:fiz064, doi: 10.1093/femsec/fiz064, 2019.

4.  In this manuscript, the "typical" cell concentration per red snow volume is mentioned several times. But what is typical? First of all, it depends on the location of the ecosystem. Coastal snow fields are usually affected by local nutrient input and may have striking blood red coloration, whereas oligotrophic alpine and polar snow has much lower abundances, causing a more pink snow.

We agree with you. Typical cell concentration for red snow depends on the location of the ecosystem. We think that the study site (Qaanaaq Glacier) fall into the category of oligotrophic polar snow. The information of the location (oligotrophic polar and alpine snow) for definition of "typical red snow" has been added in the manuscript (Line 319). In addition, the range of the typical algal cell concentration in this study has been modified (Line 320). Accordingly, the results in Figures 6 and 7 and the sentences in from Discussion 4.5 to Discussion 4.8 have been modified.

Lines from 317 to 322 in the revised manuscript:

This is probably due to the lower cell concentrations at the study site, which was $4.9 \pm 1.7 \times 10^4$ cells L$^{-1}$ (mean $\pm$ SD) on day 215, when compared with those of typical red snow appeared on oligotrophic polar or alpine snow, which range from $3.2 \times 10^6$ to $2.0 \times 10^8$ cells L$^{-1}$ (Thomas and Duval, 1995; Takeuchi and Koshima, 2004; Takeuchi et al., 2006; Stibal et al., 2007; Takeuchi, 2013; Lutz et al., 2014; ; Tanaka et al., 2016; Onuma et al., 2018; Procházková et al., 2018). In fact, visible red snow was not seen on day 215 at the study site.

5.  Moreover, the concentration of the organic impurities on the snow surface can also be influenced

by meteorological events like precipitation, and the authors should discuss this aspect.

The precipitation event should be discussed to reveal a factor affecting temporal changes in algal abundance. However, we believe that the discussion would be outside the scope of our paper because snow algal abundance did not significantly affect the surface albedo during the period. The OC and algal cell concentrations ranged from $3.2 \times 10^{-2}$ to $1.3 \times 10^{-1}$ mg $L^{-1}$ and 0 to $1.3 \times 10^{-4}$ cells $L^{-1}$ in the surface snow from day 168 to day 209, respectively, indicating that the concentrations are pretty lower than that of typical red snow in oligotrophic polar snow. The discussion about the temporal changes in the lower algal cell concentration may lead to confuse the issue (i.e. contribution of red snow blooming to snow albedo). Therefore, we did not add the discussion to the manuscript in this time although precipitation event is an important factor to discuss temporal changes in snow algal abundance.

6. Melting surface snow has temperatures about 0.5 to 1 C in all studies I know. Therefore, why were calculations performed simulating an elevated snow temperature of +1.5C? This should be physically impossible, and thus I suggest removing this hypothetical data from the manuscript, and likewise change fig. 7.

In our simulation, snow temperature becomes 0°C when the temperature exceeds 0°C by warming test. Our explanation in the manuscript was insufficient about that, so the explanation has been described (Lines from 445 to 446).

Lines from 443 to 446 in the revised manuscript:
The simulation was conducted for 30 days, starting after day 215. The temporal changes in surface snow temperature under different assumptions were used as input variables for the ==snow algae model==. The range of the surface snow temperature was within plus or minus 1.5°C. ==In the model simulation, snow temperature becomes 0°C when the temperature exceeds 0°C by warming test.==

Detail comments:
1. Line 16: use Sanguina nivaloides instead of Chlamydomonas nivalis throughoutge'
The species name has been corrected in the whole of the manuscript.

2. Line 27: Why should this albedo model only be valid for snow covers in Greenland? Could it be applied elsewhere?
The albedo model was established on the basis of the observation in Qaanaaq Glacier. Unfortunately, we have only the validation data in the study site (snow impurity concentrations and snow physical

properties in surface and subsurface snow layers). However, we plan to apply the model to the other site (for example, Alaskan glacier and Svalbard). The current issue and future plan have been described in the manuscript (Lines from 381 to 385).

3. Line 41: visible is from 400 to 700 nm
The word has been corrected (Line 42).

4. Line 47: "of" instead "in" the Greenland Ice Sheet
The word has been corrected (Line 48).

5. Line 50: insert "algae" to the list in brackets
The word has been inserted (Line 52).

6. Line 52: insert "in" after "might be present"
Following another reviewer's comment, "on" has been inserted after "present" (Line 52).

7. Line 57: Start sentence with "The …"
The word has been inserted (Line 58).

8. Line 67: reference Yallop et al. 2012 is about glacier algae, not about snow algae
The reference has been removed (Line 69).

9. Line 68: delete "and ice" for snow algae.
The word has been deleted (Line 70).

10. Line 70: give reference Hoham & Remias instead of Hoham & Duval
The reference has been replaced with Hoham & Remias (Line 71).

11. Lnie 71: use Sanguina nivaloides instead of Chlamydomonas nivalis and include reference Procházková et al 2019
The sentence has been modified as suggested. In addition, the reference has been included in the sentence (Lines from 72 to 75).

Lines from 70 to 75 in the revised manuscript:

Blooms of snow algae occur on thawing snow surfaces and change the color of snow to red or green (Thomas and Duval, 1995; Takeuchi et al., 2006; Hoham and Remias, 2020). Red colored snow results

from a bloom of snow algae, which are typically *Sanguina (S.) nivaloides* (renamed recently from *Chlamydomonas nivalis*), and can be observed widely in polar and alpine snow fields (Hoham and Duval, 2001; Segawa et al., 2005; Takeuchi, 2013; Hisakawa et al., 2015; Lutz et al., 2016; Tanaka et al., 2016; Ganey et al., 2017; Segawa et al., 2018; Procházková et al., 2019).

12. Line 74: "algal blooms have" instead of "algal blooming has"

The words have been corrected (Line 76).

13. Line 78: "cellular pigment composition", delete "in cell"

The words have been corrected (Line 80).

14. Line 84 & 93: reformulate as explained earlier: the algal abundance on surface is not due to growth but due to concentration. E.g. "accumulation model"

As we mentioned at Major comment 3, our model considers both effects of algal growth and accumulation to reproduce algal cell abundance. We have modified the sentence (Lines from 86 to 87).

Lines from 85 to 87 in the revised manuscript:

In addition, temporal changes in algal abundance were not used at the model calculation. Snow algal abundance can change significantly because of their growth, accumulation and removal of their cells over time (Müller et al., 2001; Takeuchi, 2013; Onuma et al., 2016; 2018). Therefore, snow albedo simulations should incorporate a numerical model of snow algae.

15. Line 191: The mean cell radius was 11.4 μm – based on algae from the study site? Please state. Elsewhere in the manuscript, a different cell size number based on an older study of Onuma et al. was given.

Yes, the size was measured using the cells observed in the study site. Onuma et al. (2018) showed that the mean cell size at the study sites (Sites A and B in the previous study), but we showed that the mean cell size at Site B only in this study. The explanation has been added to the manuscript (Line198).

16. Line 237: the decrease of OC on days 197 and 209 should be discussed, and please consider meteorological events like precipitation as a causer?

As we answered at Major comment 5, we did not add the discussion to the manuscript in this time because we believe that the discussion would be outside the scope of our paper.

17. Lines 240 – 245: The correlation between OC concentration and snow algae abundance appears

to be obvious. Still, despite high significance, other organisms can also contribute to OC like bacteria, and in this habitat most likely, yeast-like fungi. This should be at least briefly discussed (e.g. line 285: the formula is not a proof that snow algae are the main constitute of OC in snow – though this is apparent for this study). Since fungal blooms depend on the snow algae, there could be a stable correlation as well.

Although we could not quantify the abundance of fungi, we have quantified the abundance of *Chroococcaceae cyanobacterium* during the observational period. The abundance ranged from 0 to $9.7 \times 10^5$ cells m$^{-2}$ from day 168 to day 215, so it possibly contributes to OC concentration. However, the cell size was very smaller ($2.3 \pm 0.6$ μm in radius) compared with that of red cyst of *Sanguina nivaloides*. Therefore, we assumed that the effect of the cyanobacteria can be neglected to obtain the relationship between the red cyst abundance of *Sanguina nivaloides* and the concentration of OC. As you pointed out, we should discuss the contribution of bacteria and fungi to OC concentration. The discussion has been described (Lines from 299 to 304).

Lines from 297 to 304 in the revised manuscript:

Indeed, *S. nivaloides* was the dominant species in snowpack at the study site throughout the summer season of 2014 (Onuma et al., 2018). However, significant amounts of OC were detected in snow samples without cells of *S. nivaloides*, indicating that these snow samples contained organic matter originated from the other organisms (for example, bacteria and yeast-like fungi) and atmospheric OC aerosol. The intercept of 0.0826 of equation (3) can be interpreted as to be contributed from the other organisms and the atmospheric OC aerosol. In fact, Chroococcaceae cyanobacterium, which is a cyanobacteria found commonly on glaciers and snowpacks in Greenland, was observed on the surface snow of the study site from mid-June to early August in 2014 (Onuma et al., 2018). However, their effect was neglected in the present study because the concentration was much smaller than the abundance of *S. nivaloides* at the study site.

18. Line 285: "constituent" instead of "constitution"

The word has been corrected (line 295).

19. Line 289: please explain/hypothesise how algae-free snow contains significant amount of OC at a remote place of Northwestern Greenland. Long-distance deposition?

Following the previous your comment (Detail comments 17), we reconsidered the reason for the intercept of 0.0826 of equation (3) in the manuscript. Probably, the contribution of cyanobacteria abundances is included in the intercept value. The discussion about the reason of the intercept value has been described (lines from 299 to 304).

20. Line 296: unneeded repetition of the considered pigments

The words have been deleted (Line 307).

21. Line 297: "absorption", not "abruption"

The word has been corrected (Line 307).

22. Line 299: Sentence "The dominant species was …" is a repetition of results.

The sentence has been removed (Line 310). Also, sentence "These results suggest" has been corrected (Line 310).

23. Line 309: the typical cell concentration of the study site was 4.9 x 10 e4 cells per litre. To my experience, the lower threshold to see snow discolorations by one's eye is about 5 x 10 e6. Thus my question, did you see red snow visually at the study site? This should be mentioned in the results.

We could not see red snow visually at the study site in 2014 season. However, red snow appeared visibly on surface snow at the lower site of Qaanaaq Glacier when the cell concentration of the red cyst of *Sanguina nivaloides* was $3.2 \times 10^6$ cells L$^{-1}$ (Onuma et al., 2018). To our knowledge, the cell concentration was slightly lower than the concentration in the other sites reported by previous studies. So, we assumed that the minimum value of typical red snow is $3.2 \times 10^6$ cells L$^{-1}$ in this study. The explanation has been added (Lines from 319 to 322).

24. Line 310: delete non-existing reference "Sutton et al. 1972". There exists only a PhD of Mrs. Sutton from that year, a fine work though which was never published as a paper. There are many good papers for the concentration of Sanguina nivaloides in red snow.

The reference has been deleted, so some reference (Takeuchi and Koshima, 2004; Takeuchi et al., 2006; Stibal et al., 2007; Tanaka et al., 2016; Procházková et al., 2018) have been added in the manuscript instead (Lines from 320 to 321). The range of the typical red snow concentration has been modified (Line 320). Also, the snow albedos were recalculated using the newer range of the algal cell concentration with PBSAM. Accordingly, the results in Figures 6 and 7 and the sentences in from Discussion 4.5 to Discussion 4.8 have been modified.

Reference:

Takeuchi, N. and Kohshima, S.: snow algal community on a Patagonian glacier, Tyndall glacier in the Southern Patagonia Icefield, *Arct. Antarct. Alp. Res.*, 36, 91–8, 2004.

Takeuchi, N., Dial, R., Kohshima, S., Segawa, T. and Uetake, J.: Spatial distribution and abundance

of red snow algae on the Harding Icefield, Alaska derived from a satellite image. *Geophys. Res. Lett.*, 33, L21502, doi: 10.1029/2006GL027819, 2006.

Stibal, M., Elster, J., Ŝabacká, M. and Kaŝtovská, K.: Seasonal and diel changes in photosynthetic activity of the snow alga Chlamydomonas nivalis (Chlorophyceae) from Svalbard determined by pulse amplitude modulation fluorometry. *FEMS. Microbiol. Ecol.*, 59, 265–273, doi:10.1111/j.1574-6941.2006.00264.x, 2007.

Tanaka, S., Takeuchi, N., Miyairi, M., Fujisawa, Y., Kadota, T., Shirakawa, T., Kusaka, R., Takahashi, S., Enomoto, H., Ohata, T., Yabuki, H., Konya, K., Fedorov, A., Konstantinov, P.: Snow algal communities on glaciers in the Suntar-Khayata Mountain Range in eastern Siberia, Russia. *Polar Sci.*, 10, 3, 227-238, doi: 10.1016/j.polar.2016.03.004, 2016.

Procházková, L., Remias, D., Holzinger, A., Řezanka, T. and Nedbalová, L.: Ecophysiological and morphological comparison of two populations of Chlainomonas sp. (Chlorophyta) causing red snow on ice-covered lakes in the High Tatras and Austrian Alps., *Eur. J. Phycol.*, 53:230–43, doi:10.1080/09670262.2018.1426789, 2018.

25. Chapter 4.3 (lines 316 – 337): I have serious problems with this part. It should either be rewritten or omitted. There are wrong biological assumptions about growth of red snow algal cysts (as stated in the beginning), which simply does not take place on the snow surface. The bloom that you can see is a (physiologically active) resting stage.

In this study, cell concentration of red cyst of *S. nivaloides* was calculated with "snow algae model". As we mentioned at Major comment 3, we have described the detailed assumption and current issue of the snow algae model (Lines from 336 to 358).

26. Chapter 4.4: A "typical" red snow bloom should be defined more clearly. Only two references for average cell numbers are given. Furthermore, Painter et al 2001 is inappropriate in this context, because they clearly state that they did not any field measurements. Moreover, the number taken from Lutz et al 2014 from location MIT-17 is not the average value for red cysts in that study, but a rather low value for a bloom. For being representative, I strongly recommend to include certain further studies of "Chlamydomonas nivalis" and then recalculate the albedo reduction of a "typical" red bloom.

As we mentioned at Major comment 4, the definition of "typical red snow" has been added in the manuscript (Line 319). And, we have modified the range of "typical red snow" in the manuscript. Accordingly, the snow albedos were recalculated using the newer range of the algal cell concentration

with PBSAM. Based on your comment, we have added model validations with two field observation data (Takeuchi et al., 2006 and Takeuchi, 2013). In this study, we used only field observation data collected by Painter et al. (2001, Fig. 1). To our understanding, the data of the figure 1 seemed to be obtained from field measurement and sampling although the other data were not obtained from those. In Lutz et al. (2014), the snow albedo in the visible band and algal cell concentration were observed simultaneously at Mit-17, so we consider that the model validation can be conducted using the observational data. To explain clearly, we have summarized the information about these model validations as a table (Lines from 357 to 359 and Table 2). Based on these validations, we have discussed about possible biological factors affecting snow albedo (cell size, algal community and pigment composition) (Lines from 362 to 385).

Lines from 357 to 385 in the revised manuscript:

We validated the albedo reduction for high algal abundance using the snow albedos of red snow surface on oligotrophic polar or alpine snow reported by Previous studies (Painter et al., 2001; Takeuchi et al., 2006; Takeuchi, 2013; Lutz et al., 2014) (Table 2). The algal cell concentrations obtained from their field measurements were used as input variables in surface (0–2 cm) and subsurface (2–10 cm) snow. These algal cell concentrations were converted into $C_{OC}$ using equation (3). Our observational data on day 215 (meteorological, snow physical and impurity conditions) were used as other input data of these simulations. The simulation using the cell concentration observed by Painter et al. (2001) demonstrated that the difference between Alb-DB and Alb-DBA was 0.062, which is equivalent to the albedo reduction by snow algae, and in agreement with the algal albedo reduction (0.07) observed by Painter et al. (2001). This reduction in albedo was also close to the result of another simulation with the bio-albedo model proposed by Cook et al. (2017a, algal albedo reduction = 0.07). Thus, both our PBSAM and the bio-albedo model can consistently reproduce the reduction in albedo based on the optical properties of *S. nivaloides*. The simulation using the cell concentration observed by Takeuchi (2013) suggested that the simulated albedo reduction closed to the observed that (model: 0.105, observation: 0.12). In contrast, the simulation using the cell concentration reported by Lutz et al. (2014) produced an albedo reduction by snow algae of 0.015, which was lower than that observed by them (0.09) and calculated with the bio-albedo model (0.09). This is probably owing to different algal pigments in the ice surfaces. Lutz et al. (2014) reported that glacier algae (filamentous cells: $6.1 \times 10^6$ cells $L^{-1}$) were found in addition to snow algae (spherical cells: $1.8 \times 10^6$ cells $L^{-1}$) in the samples collected at their study site (MIT-17). The phenolic pigments of glacier algae have a broader bandwidth of spectral absorption than the carotenoids and chlorophyll of *S. nivaloides* (Remias, 2012; Williamson et al., 2020). In the albedo simulation with the bio-albedo model, measured pigment compositions (total chlorophyll, primary and secondary carotenoids) were used as model parameters while our simulation only used MAC for snow algae (*S. nivaloides*). The simulation using the cell concentration

reported by Takeuchi et al. (2006) showed that the simulated albedo reduction underestimated the observed albedo reduction (model: 0.072, observation: 0.099). This may be due to the difference between the observed and parameterized cell size. Our PBSAM assumed that the cell size of *S. nivaloides* is 11.4 μm, whereas that measured by Takeuchi et al. (2006) was 17.5 μm. Because the MACs for red snow were estimated using the cell size of 11.4 μm, the simulated mass absorption might underestimate the intact mass absorption for red snow algae. Unfortunately, we have only the validation data in the study site (MD, BC and OC concentrations and snow physical properties in surface and subsurface snow layers). The detailed time series observation, including analysis of cell size, pigment composition, algal community, should be conducted in other sites to evaluate our albedo model. Moreover, the detailed spatial measurements of algal cell abundance and snow albedo would also be needed because patchy distribution of red snow often appear on oligotrophic polar and alpine snow.

27. Line 353: Start sentence "The phenolic pigments of glacier algae have a broader bandwidth of spectral absorption than the carotenoids of S. nivaloides". Instead of citing Dial et al. 2018, which is rather a deductive/theoretical work, a reference showing real spectral data is more appropriate (e.g. Fig. 13.6 in Remias 2012, Springer Vienna)

The sentence has been modified as suggested (lines from 372 to 374). And, the following references have been added in the manuscript.

Reference:
Remias, D.: Cell structure and physiology of alpine snow and ice algae, in: Plants in alpine regions, Cell physiology of adaption and survival strategies, edited by: Lütz, C., *Springer Wien*, 202, 175–186, doi: 10.1007/978-3-7091-0136-0_13, 2012.

Williamson, C. J., Cook, J. M., Tedstone, A., Yallop, M., McCutcheon, J., Ponieckae, E., Campbell, D., Irvine-Fynn, T., McQuoid, J., Tranter, M., Perkinse, R. and Anesio, A.: Algal photophysiology drives darkening and melt of the Greenland Ice Sheet, *PNAS*, 117, 11, 5694-5705, doi:10.1073/pnas.1918412117, 2020.

28. Line 363: Again, Sutton 1972 is not a paper and Painter et al 2001 give no numbers to cite for.

These reference have been deleted, so some reference (Takeuchi and Koshima, 2004; Takeuchi et al., 2006; Stibal et al., 2007; Tanaka et al., 2016; Procházková et al., 2018) have been added in the manuscript instead (Lines from 391 to 392).

29. Line 388: replace "but" with "and"

The word has been corrected (Line 417).

30. Chapter 4.8: Needs to be updated. Please think over the "snow algae growth model" and the sense of using hypothetical high snow temperatures for any calculations.

As we mentioned at Major comments 3, these explanations have been added to the discussion part in the manuscript (Lines from 328 to 355).

31. Growth of snow algae takes place in deep layers of snow with very constant conditions around 0.5°C. The air temperature plays generally no big role since the snow pack starts to be water logged from the bottom, and this occurs earlier than air temperatures raise above zero.

We also think that melting water is an important factor affecting algal growth. Onuma et al. (2018) reported that cells of *S. nivaloides* were not observed from bottom to surface in snowpack at this study site on mid-June in 2014. This is probably due to existence of super imposed ice on the bottom layer. In the case, the red algal cells appear to have originated from windblown algal spores in the atmosphere, but they are not likely from the remaining snow of the previous melt season. Moreover, snow surface temperature was almost 0°C at the study site from late June to early August in 2014. Logistic model may reproduce the temporal change in algal abundance under such conditions. Because the explanation has been described in Onuma et al. (2018), the brief explanation has been added to our manuscript as suggested (Lines from 328 to 355).

32. Line 586: Reference Painter et al 2001 incomplete, authors missing. "Thomas" instead of Thimas".

The author name has been corrected (Line 622).

33. Legend to fig 1: "sampling site", not "sites"

The word has been corrected (Fig. 1).

34. Legend to fig. 5: say "Temporal changes in algal cell concentration and …"

The words have been replaced as suggested (Fig 5).

35. Figure 7: This is a bit vague considering the theory of cell growth and snow temperatures of +1.5°C

In our snow algae model, snow algal cell concentration gradually increased with snow melting, which means that snow temperature on surface exceed 0°C. And, snow temperature becomes 0°C when the temperature exceeds 0°C by the warming test. Therefore, the sensitivity test of temporal changes in the algal cell abundance can be conducted with the snow algae model using different snow temperature conditions. The explanation has been described (Lines from 445 to 446).

---

## Author Response (AR1)

May 11, 2020

Dr. Yukihiko Onuma

Institute of Industrial Science, University of Tokyo

E-mail: onuma@iis.u-tokyo.ac.jp

Dr. Elizabeth Bagshaw

Editor, The Cryosphere

Dear Dr. Elizabeth Bagshaw,

We appreciate the time and effort you have dedicated to providing insightful feedback on ways to strengthen our paper. Please see enclosed our responses to the all your comments. Our responses (**blue text**) to each your comment (**black text**) were described on the following pages. We also described major revised sentences based on your suggestion (revised part: yellow marker), after our response. In addition, we have uploaded the revised marked-up manuscript entitled as "Physically-based model of the contribution of red snow algal cells to temporal changes in albedo in northwest Greenland" by Yukihiko Onuma et al. [Paper # tc-2019-263]. The manuscript reflects comments from you and two referees (revised part: yellow marker).

Best regards,

Yukihiko Onuma and co-authors

**Editors report on TC-2019-263, Onuma et al.**

Thank you for your thorough response to the two reviewers. I now invite you to incorporate these changes into a revised manuscript. In addition to the changes requested by the reviewers, I also request that you consider my suggestions below, which are based on my assessment of the reviews and the response. I look forward to receiving the revised version of the manuscript.

Kind regards,

Dr Elizabeth Bagshaw

Title: I suggest shortening this amended title, perhaps to 'Physically-based model of the contribution of snow algal cells to temporal changes in albedo' or 'Physically-based modelling of algal-induced snow albedo changes' or similar. The new title is, of course, fully correct and addresses the reviewer's concerns, but I find it overly long-winded.

We have corrected the title as suggested. The title is as below.

"Physically-based model of the contribution of red snow algal cells to temporal changes in albedo in northwest Greenland"

L46: you suggest that 'to explain clearly, the value (0.7%) has been changed to 'approximately 1%' (Line 47).' I instead suggest simply rewriting to 'was reported to be 0.7% lower than BC'.

The value has been corrected (Line 47).

L103: Suggest adding a qualifier to this sentence to acknowledge the reviewer's suggestion: 'They consisted mostly of the spherical red cells of Cd. nivalis, with mean diameter was 21.3 ± 2.3 µm (Onuma et al., 2018), although there was no molecular analysis of all species present'.

The sentence has been added to the manuscript (Line 107).

L328-343 in revised manuscript: 'activity' should be 'actively' in 'and thus not activity increase their population.'

The word has been changed (Line 339).

L351 New citation: Williamson et al. 2020. Please check your citation, since there are several typos in the author list you include in your response.

These typos have been corrected in the reference list (Line 701).

L381-385 in revised manuscript: please rephrase this final sentence: 'Moreover, the detailed spatial measurements of algal cell abundance and snow albedo would also be needed because patchy

distribution of red snow often appear on oligotrophic polar and alpine snow.' – patchy snow appears on snow?

The finial sentence has been rephrased (Lines from 385 to 386).

L426: 'accelerated loss in mass balance' is not strictly correct. I would rephrase to 'accelerated loss in mass' or 'decreased mass balance' or similar.

The sentence has been corrected as suggested (Lines from 457 to 458).

L443-446 in revised manuscript: 'In the model simulation, snow temperature becomes 0°C when the temperature exceeds 0°C by warming test.' I am unclear what a 'warming test' is, please rephrase.

We have revised the sentence as suggested (Lines from 444 to 445). Also, 'scenario simulation' has been changed to 'sensitivity test' (Line 437).

Table 1: please rewrite these values in scientific notation, eg. 8 x 10-3

Those values have been rewritten in Table1 (for example, 1.5E-01 -> $1.5 \times 10^{-1}$).

Figure 2: please label the x-axis in at least the bottom figure

The x-axis label (Day of the year (2014)) has been added into the Figure 2.

Figure 7: first, be consistent with 'Fig, 7' or 'Fig. 7' in the text. Second, I find the justification for the gray shading hard to understand, even with your explanation in response to the reviewer. Please can you very clearly state what the shading indicates, and how/why this differs from the coloured lines on the plot?

'Fig, 7' was used consistently throughout the text (Line 451).

We have deleted the gray shading, but used a simple red line instead, to show the simulated result of the cases of plus and minus 1.5 °C in Figure 7. The text in Section 4.8 has been revised to explain the simulation clearly. The revised paragraph and modified figure are as below.

Lines from 441 to 450 in the revised manuscript:

[revised manuscript text omitted]

---

## Editor Decision (ED1)

Thank you for your thorough response to the two reviewers. I now invite you to incorporate these changes into a revised manuscript. In addition to the changes requested by the reviewers, I also request that you consider my suggestions below, which are based on my assessment of the reviews and the response.

I look forward to receiving the revised version of the manuscript.

Kind regards,

Dr Elizabeth Bagshaw

Title: I suggest shortening this amended title, perhaps to 'Physically-based model of the contribution of snow algal cells to temporal changes in albedo' or 'Physically-based modelling of algal-induced snow albedo changes' or similar. The new title is, of course, fully correct and addresses the reviewer's concerns, but I find it overly long-winded.

L46: you suggest that 'to explain clearly, the value (0.7%) has been changed to 'approximately 1%' (Line 47).' I instead suggest simply rewriting to 'was reported to be 0.7% lower than BC'.

L103: Suggest adding a qualifier to this sentence to acknowledge the reviewer's suggestion: 'They consisted mostly of the spherical red cells of Cd. nivalis, with mean diameter was 21.3 ± 2.3 µm (Onuma et al., 2018), although there was no molecular analysis of all species present'.

L328-343 in revised manuscript: 'activity' should be 'actively'  in 'and thus not activity increase their population.'

L351 New citation: Williamson et al. 2020. Please check your citation, since there are several typos in the author list you include in your response.

L381-385 in revised manuscript: please rephrase this final sentence: 'Moreover, the detailed spatial measurements of algal cell abundance and snow albedo would also be needed because patchy distribution of red snow often appear on oligotrophic polar and alpine snow.' – patchy snow appears on snow?

L426: 'accelerated loss in mass balance' is not strictly correct. I would rephrase to 'accelerated loss in mass' or 'decreased mass balance' or similar.

L443-446 in revised manuscript: 'In the model simulation, snow temperature becomes 0°C when the temperature exceeds 0°C by warming test.' I am unclear what a 'warming test' is, please rephrase.

Table 1: please rewrite these values in scientific notation, eg. $8 \times 10^{-3}$

Figure 2: please label the x-axis in at least the bottom figure

Figure 7: first, be consistent with 'Fig, 7' or 'Fig. 7' in the text. Second, I find the justification for the gray shading hard to understand, even with your explanation in response to the reviewer. Please can you very clearly state what the shading indicates, and how/why this differs from the coloured lines on the plot?